# Responsive core-shell DNA particles trigger lipid-membrane disruption and bacteria entrapment

Michal Walczak [1], Ryan A. Brady[2], Leonardo Mancini[1], Claudia Contini [3,4], Roger Rubio-Sánchez [1], William T. Kaufhold[1,3], Pietro Cicuta [1] & Lorenzo Di Michele [1,3,4 ✉]

Biology has evolved a variety of agents capable of permeabilizing and disrupting lipid membranes, from amyloid aggregates, to antimicrobial peptides, to venom compounds. While often associated with disease or toxicity, these agents are also central to many bio-sensing and therapeutic technologies. Here, we introduce a class of synthetic, DNA-based particles capable of disrupting lipid membranes. The particles have finely programmable size, and self-assemble from all-DNA and cholesterol-DNA nanostructures, the latter forming a membrane-adhesive core and the former a protective hydrophilic corona. We show that the corona can be selectively displaced with a molecular cue, exposing the 'sticky' core. Unprotected particles adhere to synthetic lipid vesicles, which in turn enhances membrane permeability and leads to vesicle collapse. Furthermore, particle-particle coalescence leads to the formation of gel-like DNA aggregates that envelop surviving vesicles. This response is reminiscent of pathogen immobilisation through immune cells secretion of DNA networks, as we demonstrate by trapping *E. coli* bacteria.

[1] Biological and Soft Systems, Cavendish Laboratory, University of Cambridge, JJ Thomson Avenue, Cambridge, UK. [2] Department of Chemistry, Faculty of Natural and Mathematical Sciences, King's College London, London, UK. [3] Department of Chemistry, Molecular Sciences Research Hub, Imperial College London, London, UK. [4] fabriCELL, Molecular Sciences Research Hub, Imperial College London, London, UK. ✉email: l.di-michele@imperial.ac.uk

Proteolipid membranes represent the main means through which biological cells sustain chemical heterogeneity at the micro- and nanoscale, enabling most of their astounding responses. Several biomolecular agents, from small molecules to large protein complexes, have evolved the ability to sculpt lipid membranes changing their morphology, chemical composition, and physical properties. Some of these agents are central to physiological processes[1], including protein channels mediating molecular transport[2,3], enzymes dynamically regulating lipid composition and distribution[4,5], and protein complexes that establish local membrane curvature to promote endo/exocytosis or cell division[6–9]. In other cases, membrane manipulation is associated with pathology, for instance in the action of viruses and parasites hijacking membrane-trafficking machinery to enter/ leave cells[10,11], pore-forming toxins permeabilizing membranes[12], and neurotoxic protein aggregates[13].

The action of membrane-destabilizing biological agents is central to several biosensing, diagnostic, therapeutic, and synthetic-biological platforms. Notable examples include the routine adoption of $\alpha$-hemolysin and other nanopore-forming proteins in single-molecule sensing and nucleic-acid sequencing[14,15], and membrane-piercing antimicrobial peptides[16].

In view of this applicative potential, efforts have been devoted to engineering natural membrane-sculpting entities[15,17], and even mimic their responses with purely synthetic analogues. DNA nanotechnology, with its yet unparalleled ability to construct nanoscale motifs of near-arbitrary shape and responsiveness, offers an ideal toolkit for mimicking the complex responses of membrane-active biological machinery[18,19]. Simple amphiphilic DNA nanostructures have been utilized to engineer membrane adhesion and the formation of artificial tissues in both synthetic membranes[20–23] and cells[24]. Trans-membrane DNA constructs can mimic biological pores[25,26] and simple enzymes[27], while membrane-adhering DNA-origami have been shown to establish local membrane curvature[28,29].

Here we demonstrate the design of responsive, DNA-based particles capable of permeabilizing and disrupting model lipid membranes when triggered by a molecular cue. The particles self-assemble in a one-pot reaction from cholesterol-modified and plain synthetic oligonucleotides, designed to form three distinct motifs. One of these nanostuctures, with an amphiphilic character, forms the core of the particles, while the others form a stabilizing "corona". This protective layer suppresses particle–membrane interactions while, combined with a tailored thermal annealing protocol, also enables precise control over particle size.

The corona can be removed via toehold-mediated strand displacement[30], exposing the amphiphilic core. Unprotected particles readily aggregate with each other and deposit onto nearby membranes, leading to their enhanced permeability and rupture.

The membrane-disrupting DNA particles mimic the action of biological toxins and could form the basis of smart therapeutic platforms, where target cells or pathogens are locally damaged if a molecular trigger is present.

At later aggregation stages, the particles form a network-like structure that embeds surviving liposomes, and is reminiscent of the Neutrophil Extracellular Traps, or "DNA NETs" secreted by immune cells to capture pathogens[31–34]. To explore this analogy we demonstrate that aggregating DNA particles can capture and immobilize motile Escherichia coli (E. coli).

## Results

### Self-assembly of core–shell DNA particles with prescribed size.
Figure 1a shows the three DNA motifs from which the core–shell particles are formed. The core motifs are cholesterol-modified

four-arm nanostars, or "C-stars". We have recently introduced these amphiphilic nanostructures and explored their unique tendency to self-assemble into macroscopic single crystals with programmable structure and stimuli-responsiveness[35–37]. In the present contribution, however, C-stars are not exploited for their prowess to crystallize (although we still observe crystals), but rather for the exquisite control one can afford on their self-assembly and the ability of C-star aggregates to disrupt lipid membranes. As with previous designs, here each C-star assembles from four different oligonucleotides, making up the central junction, and four identical cholesterol-modified strands[36]. Core motifs feature two pairs of single-stranded (ss)DNA overhangs along the arms, two with domain sequence labeled as $\alpha^*$ and two with sequence $\gamma$. The former mediate the interactions with the corona, while the latter are included as potential anchoring points for molecular cargoes.

The two nanostructures forming the particle shell (corona) are both six-pointed DNA nanostars, labeled as "inner" and "outer corona motif". All six arms of the inner corona motifs end with ssDNA overhangs. One of the overhangs features sequence $\alpha$, complementary to domain $\alpha^*$ on the core motifs, and mediates binding between the latter and the inner corona motifs. The remaining five overhangs have sequence $\beta^*$. The outer corona motifs only host one ssDNA overhang of sequence $\beta$, complementary to $\beta^*$, while all other arms end with blunt duplexes.

The sequences of all involved oligonucleotides are reported in Supplementary Table 1, while Figs. S1, S2 demonstrate their correct assembly and mutual interactions by means of gel electrophoresis and Dynamic Light Scattering (DLS).

The self-assembly pathway of protected DNA particles of prescribed size is sketched in Fig. 1b. All ssDNA components of all motifs are included at once in stoichiometric ratios, namely, such that the molar ratio of assembled core, inner corona, and outer corona motifs is 1:2:10, and each motif can potentially connect to others leaving no overhangs unbound. We shall see below that this is, in fact, unlikely to happen. The molar composition of the samples is reported in Supplementary Table 2.

Samples are initially heated up to 90 °C, in which conditions no double-stranded (ds)DNA is present, and ssDNA components coexist with cholesterol-DNA micelles[36] (Fig. 1b, left). A rapid temperature quench is then performed to 65 °C. As previously reported,[36] upon cooling below a threshold temperature (in the present case exceeding 65 °C) phase separation occurs in C-star samples as the dsDNA junctions start to cross-link the cholesterol-DNA micelles, leading to the nucleation and growth of C-star particles (Fig. 1b, center).

At $T = 65$ °C also inner and outer corona motifs are largely formed, but the length and sequence of $\alpha-\alpha^*$ and $\beta-\beta^*$ domains are designed so that their hybridization does not take place at this temperature, and corona motifs remain detached from the growing C-star particles. The characterization of the melting temperatures of the motifs, related to the formation temperature of the C-star particles, and those of the $\alpha$ and $\beta$ overhangs is discussed in Supplementary Fig. 3.

While incubated at 65 °C, C-star particles continue to grow and coalesce, and if given sufficient time can reach tens of microns in size[36]. Aggregates of these dimensions would display negligible Brownian motion and be unsuitable as membrane-disrupting agents.

After allowing for the particles to incubate at 65 °C for a growth time $t_g$, we perform a second fast quench to $T = 35$ °C (Fig. 1b, right). This temperature is sufficiently low for the $\alpha-\alpha^*$ and $\beta-\beta^*$ duplexes to assemble, hence triggering the formation of the two-layer corona around the C-star particles. The inner corona motifs are expected to bind the accessible $\alpha^*$ overhangs on the outer layers of the particles, with each inner corona motif

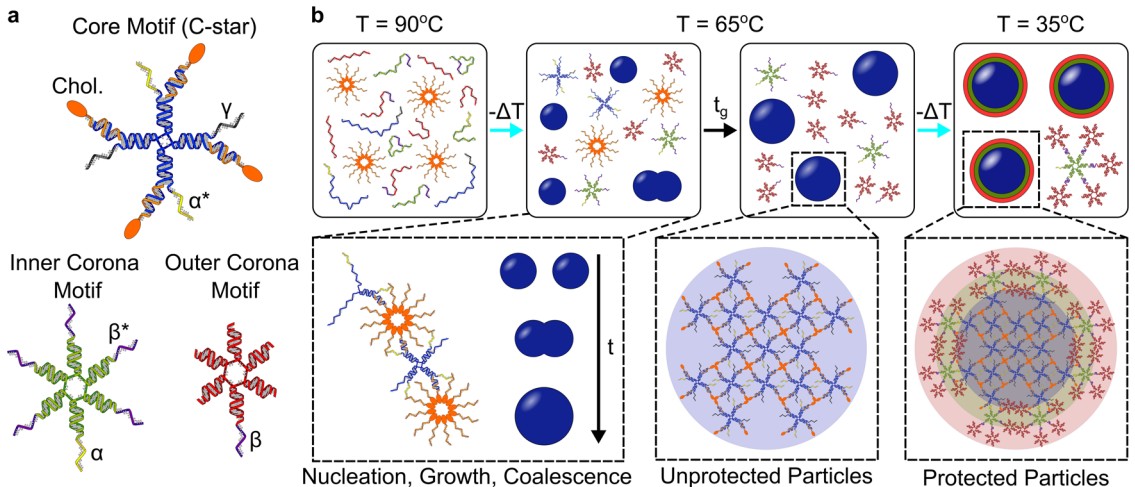

**Fig. 1 Design and assembly of particle-forming DNA nanostructures. a** DNA nanostars used in the assembly of stable, size-controlled particles. Core motifs (C-stars) assemble from four different strands forming the central junction (blue) and four identical cholesterol-functionalized oligonucleotides (orange). Inner corona motifs (green) and outer corona motifs (red) each self-assemble from six different oligonucleotides. Inner corona motifs bind to the core motifs through $\alpha-\alpha^*$ overhangs, while outer corona motifs bind to inner corona motifs through domains $\beta-\beta^*$. In the tested design, domain $\gamma$ is a non-interacting poly-T domain, which could however be replaced with a functional moiety or aptamer useful for anchoring molecular cargoes without impacting C-star self-assembly, as demonstrated in ref. [35]. **b** Particle assembly pathway. All DNA strands are mixed in stoichiometric ratios. At high temperatures ($T = 90\,°C$) cholesterol-functionalized strands form micelles while the remaining oligonucleotides freely diffuse in solution. Upon fast quenching to $T = 65\,°C$, C-stars nucleate and grow as previously reported[35-37], leading to the formation of amphiphilic DNA particles whose size depends on the incubation (or growth) time $t_g$ at $T = 65\,°C$. At this stage, individual corona motifs assemble but remain detached from the particles. When rapidly cooled down to $T = 35\,°C$, corona motifs coat the particle with a two-layer hydrophilic shell, which offers steric stabilization and prevents further coalescence.

decorated by multiple outer corona motifs, forming dendrimeric DNA constructs that surround the amphiphilic cores. The result is a dense hydrophilic brush that stabilizes the now core–shell DNA particles against further coalescence. Note that a significant fraction of the $\alpha^*$ overhangs is likely to be distant from the particle surface and thus inaccessible to corona constructs. Consequently, excess corona constructs remain free in solution (see "Methods", Fig. 1b and Supplementary Fig. 4).

The two-step self-assembly protocol we introduce enables precise control over particles size, as demonstrated with differential dynamic microscopy (DDM[38,39]) in Fig. 2a. By simply tuning the growth time, the mean hydrodynamic radius $R_H$ of the particles increases monotonically, from ~200 nm at $t_g = 15$ s to ~1.2 $\mu$m at $t_g = 1800$ s. The particle size dependency on $t_g$ is well described by a standard diffusion-reaction growth model, and discussed in Supplementary Discussion 1[40]. Cumulant analysis of the DDM data reveals that particles ($t_g = 1800$ s) are relatively polydisperse, with a polydispersity index of ~0.5, as shown in Supplementary Fig. 5 and Supplementary Discussion 2[41]. Hydrodynamic sizes comparable to those measured with DDM are detected with DLS for the tested $t_g$ range (Supplementary Fig. 6).

Once the protective corona is in place, the particles remain stable at room temperature for extended periods of time. Indeed, Fig. 2b shows a limited increase in $R_H$ within 1 h of particle self-assembly, which is found to plateau at longer timescales (Supplementary Fig. 7). This slight post-assembly growth could be a consequence of a small degree of further particle coalescence or, for larger sizes, equilibration of the barometric particle distribution. These effects bear no impact on the trends observed in Fig. 2a, as demonstrated by the data in Supplementary Fig. 8 in which a similar $t_g$-dependence is observed for $R_H$ values time-averaged over 1 h post-assembly. Notably, particle samples can be stored long-term at room temperature, and retain their colloidal stability for at least 14 days post-assembly (Supplementary Fig. 7).

Particles prepared at various $t_g$ have been imaged with TEM, and found to possess a roughly spherical morphology (Fig. 2c) and sizes aligned with those determined by DDM (Supplementary Fig. 9).

For sufficiently extended growth times, particles reach tens of microns in size (see "Methods"). In these conditions, fluorescent labeling of the core and outer-corona motifs enables a direct verification of the sought core–shell architecture with confocal microscopy, as demonstrated in Fig. 2d. Supplementary Fig. 4 demonstrates the presence of excess corona motifs in solution after completion of the two-step self-assembly protocol. These can be removed via centrifugation and washing (Fig. 2d and Supplementary Fig. 4).

If a very slow annealing is performed, similar to the one we developed for C-star crystallization[36], also the core-motifs used here are found to self-assemble into macroscopic single crystals, which however also feature a clear corona (Fig. 2e).

**Triggered corona displacement and particle destabilization.** With the motif architectures discussed to this point, the protective corona surrounding the particles can only be selectively displaced upon temperature increase. However, a simple design update, summarized in Fig. 3a, enables the isothermal removal of the corona with the addition of a molecular agent. Specifically, the overhangs on the inner corona motifs, originally only featuring domain $\alpha$ (Fig. 1a), are extended to include the 6 base-pair domain $\delta$. The latter acts as a toehold for a trigger oligonucleotide with domains $\delta^*\alpha^*$, which can then displace the $\alpha-\alpha^*$ bonds between the core and inner corona motifs, resulting in the release of the protective layers. Supplementary Figs. 10, 11 demonstrate the triggered detachment of freely diffusing corona and core motifs with AGE and DLS, while corona displacement on a large particle is visible on the confocal micrographs in Fig. 3b.

The displacement of the corona exposes the amphiphilic C-star cores, resuming particle coalescence and aggregate growth, as demonstrated in Fig. 3b. Here, upon trigger addition, DDM detects a clear increase in mean hydrodynamic size. Aggregation is visible in both bright field and fluorescence microscopy, as is the formation of branched gel-like particle aggregates at later times (Fig. 3b, top). These dense structures sink to the bottom of

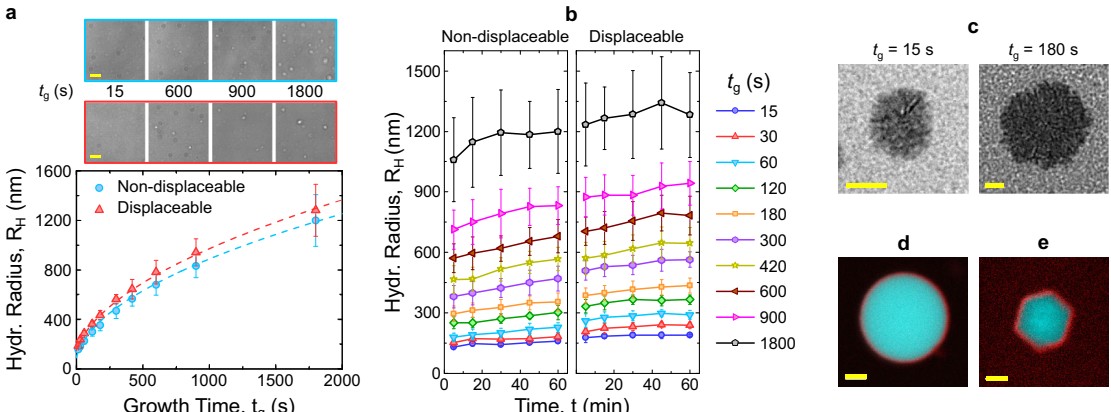

**Fig. 2 Multi-stage annealing protocol produces stable particles of programmable size. a** Hydrodynamic radius $R_H$ of protected particles with displaceable and non-displaceable corona (see Fig. 3a) as measured as a function of growth time $t_g$ using differential dynamic microscopy (DDM) (Bottom). Particle size can be prescribed by tuning $t_g$. Data are shown as mean ± standard deviation calculated over 3 (non-displaceable) and 4 (displaceable) independent repeats, each point representing the $R_H$ value depicted in panel (**b**) at the delay time of 60 min. Dashed lines are fits to a diffusion-reaction growth model (see Supplementary Discussion 1). (Top) Bright-field snapshots from the videos used in the DDM analysis, showing visibly larger aggregates for increasing $t_g$. Contrast has been enhanced to enable visualization. Scale bars 5 $\mu$m. **b** Time dependence of $R_H$ in protected particles with displaceable and non-displaceable corona as measured with DDM at room temperature. Data are shown as mean ± standard deviation as in panel (**a**). A limited increase in size is observed, demonstrating particle stability against coalescence. Supplementary Fig. 7 proves longer-term particle stability. **c** TEM micrographs of protected particles assembled at two different $t_g$ values. Selected micrographs represent data obtained in a single experiment. Scale bars 200 nm. **d** Confocal micrograph of a large aggregate assembled via a slow quenching protocol (see Methods), highlighting the core–shell structure. Scale bar 2 $\mu$m. **e** Confocal micrograph of a core–shell aggregate displaying polyhedral morphology, indicative of an underlying crystalline structure of the aggregate core, as previously demonstrated with other C-star designs[35-37]. Crystallization was achieved through slow cooling at $-0.01$ °C min$^{-1}$ (see Methods). Scale bar 2 $\mu$m. In (**d**) and (**e**) core motifs are labeled with fluorescein (cyan) and outer corona motifs with Alexa Fluor 647 (red). For both (**d**) and (**e**) image acquisition was performed twice independently.

the experimental chamber as they grow, allowing us to track particle aggregation by monitoring fluorescence intensity on that plane. Indeed, Fig. 3b demonstrates a smooth monotonic increase in mean fluorescence intensity after trigger addition.

**Triggered membrane disruption and permeabilization.** If the protective corona is displaced from small (diffusive) particles in the presence of lipid-bilayer membranes, the exposed hydrophobic moieties of the C-star cores produce an attractive interaction with membranes, as sketched in Fig. 4a. Confocal micrographs in Fig. 4b demonstrate that while protected particles have no detectable affinity for the surface of DOPC Giant Unilamellar lipid Vesicles (GUVs), exposure to the trigger oligonucleotide causes large-scale aggregation on the membrane, as well as in the surrounding bulk space (as noted in Fig. 3b).

Particle accumulation on the membranes is found to have a significant destabilizing effect on the GUVs, often leading to their bursting or collapse. A typical GUV disruption event, following triggered membrane–particle interaction, is shown in Fig. 4c (top). Here, as particles accumulate on the membrane surface and aggregates grow larger, the initial bilayer architecture is compromised and the lipids start mixing with the amphiphilic DNA aggregates. This process initially leads to shrinkage of the GUV, likely due to embedding of the membrane material in the DNA aggregates, and ultimately to its complete collapse. In some situations, GUVs become embedded in a gel-like networks of DNA particles, before possibly being disrupted (Supplementary Fig. 12a).

In other cases, as shown in Supplementary Fig. 12b, GUV collapse occurs earlier in the particle-aggregation transient, indicating that limited particle–membrane interactions may be sufficient to destabilize some GUVs. Polydispersity in membrane tension could be a discriminating factor between more fragile and resilient vesicles. In Fig. 4c (bottom, left) we exposed the vesicles, for an extended period of time, to both protected and unprotected

particles, and tracked the number of GUVs preserving their structural integrity, i.e., those that do not burst or collapse. While the presence of protected particles causes the destabilization of a small fraction (~20%) of the otherwise stable vesicles, the number of surviving GUVs drops substantially more if the protective corona is displaced. For samples with both protected and unprotected particles, GUV disruption occurs rapidly at the beginning of the experiment, before the number of "surviving" vesicles reaches a plateau. In the case of unprotected particles, the observed trend is readily explained: after the initial transient in which massive particle accumulation on the membranes takes place, the formation of large non-diffusive aggregates depletes the number of diffusive particles capable of targeting still intact GUVs. Therefore, vesicles that happen to remain intact at this stage are no longer at risk. This hypothesis is confirmed by Fig. 4c (bottom, right), where GUV "survival curves" are compared for different concentrations of unprotected particles. Here the asymptotic fraction of surviving vesicles decreases monotonically with increasing C-star concentration, consistent with the picture that more GUVs can be disrupted if more particles are present at early times, before the process is arrested by large-scale particle aggregation. Consistent with this picture, Supplementary Fig. 13 demonstrates that larger particles, expected to rapidly form sedimenting and slowly diffusing aggregates upon trigger addition, are less efficient in disrupting GUVs compared to the smaller particles examined in Fig. 4c.

Supplementary Fig. 14 shows the dependence of the asymptotic fraction of disrupted GUVs on DNA/lipid concentration ratio. Interestingly, the trend is well fitted by the Hill equation, typically used to describe dose-dependent toxicity in live organisms[42].

The rapid saturation of the fraction of surviving GUVs in the sample with protected particles is more puzzling (Fig. 4c, bottom left), as after the plateau is reached virtually all the particles remain stable in solution, and no large-scale particle aggregation is detected. The observed behavior could indicate a degree of

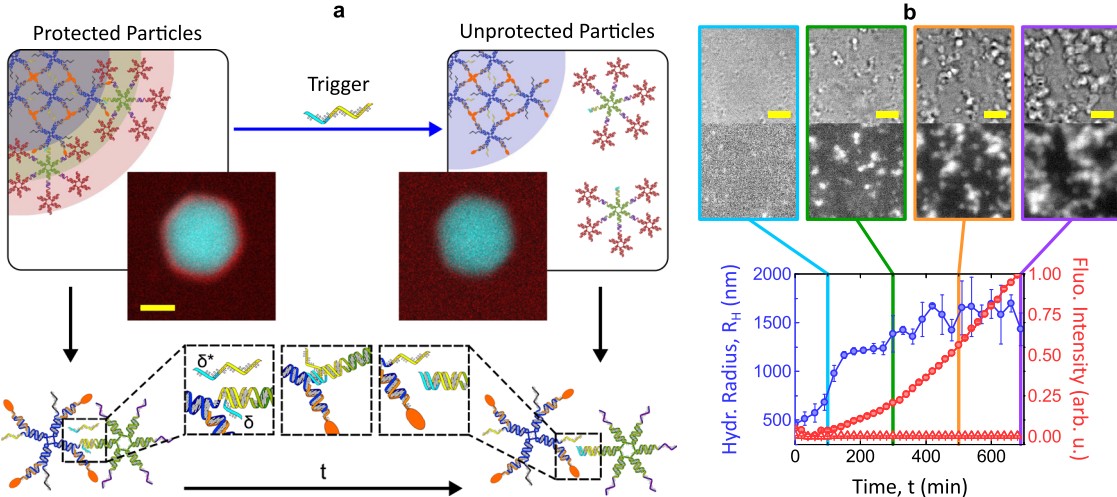

**Fig. 3 Triggered release of protective corona leads to particle aggregation. a** Schematic representation of the toehold-mediated strand displacement mechanism leading to isothermal release of the protective corona. A trigger strand of sequence $\delta^*\alpha^*$ binds the toehold domain $\delta$ on the inner corona motifs, displacing the $\alpha-\alpha^*$ bond between the latter and the core motif, and disrupting the protective shell. The confocal micrographs show a polyhedral core–shell particle before (left) and 1 min after trigger addition (right). Core motifs are labeled with fluorescein (cyan) and outer corona motifs with Alexa Fluor 647 (red). Note the increase in the background fluorescence from free corona motifs after the corona displacement. Scale bar 2 $\mu$m. **b** Corona displacement leads to the exposure of the "sticky" C-star core and subsequent particle aggregation, assessed by measuring the time-dependent hydrodynamic radius as determined via DDM (blue circles) and the normalized fluorescence intensity of labeled core motifs (red circles) after the addition of trigger strands ($t = 0$). The increase in $R_H$ observed upon trigger addition follows from the formation of larger aggregates, while the increase in the fluorescence trace is caused by their progressive sedimentation at the bottom of the cell, where the signal is recorded. For both observables, data are shown as mean ± standard deviation of 3 independent repeats. Red triangles indicate a control fluorescent trace measured in the absence of trigger. The constant and low value confirms the absence of spontaneous sedimentation. Top: bright field and fluorescence micrographs at different time-points after the addition of the trigger ($t = 100, 300, 500,$ and 690 min). All scale bars 25 $\mu$m.

heterogeneity in particle shielding. Specifically, a small fraction of the particles could be less well protected than others and able to disrupt the vesicles at early stages.

Before triggering GUV collapse, and even in the absence of any visible structural disruption, unprotected amphiphilic DNA particles enhance the permeability of DOPC membranes. To assess this phenomenon, we track the progressive release of fluorescein-sodium from DOPC GUVs, as exemplified in Fig. 4d (top). The fluorescein emission recorded in GUVs exposed to protected particles decreases at a rate very similar to that observed in the absence of any DNA, probably driven by a slow natural leakage rate and photobleaching. In turn, corona displacement leads to a substantially higher rate of fluorescence decrease, ascribed to enhanced membrane leakage (Fig. 4d, bottom left). The same trends are observed in bulk fluorimetry experiments performed on Large Unilamellar Vesicles and exploiting the self-quenching properties of concentrated calcein solutions (Supplementary Fig. 15)[43,44].

Similar to what we observed for the ability of unprotected particles to destabilize GUVs, also permeabilization is enhanced if particle concentration increases, as summarized in Fig. 4d (bottom, right).

**Bacteria capture**. The ability of the DNA particles to form a sticky web capable of surrounding cell-sized GUVs is reminiscent of the action of innate-immune cells, which can eject their genetic material to create a "DNA NET" (Neutrophil Extracellular Trap) able to entrap pathogens[31–34].

To test this potential functionality we performed experiments, summarized in Fig. 5 and sketched in panel **a**, in which motile *E. coli* are exposed to DNA particles in the presence and absence of the trigger. Confocal micrographs in Fig. 5b demonstrate that while *E. coli* (red) retain a largely uniform distribution when exposed to un-triggered particles, they become embedded in the amphiphilic DNA web formed upon trigger addition (cyan).

To quantify the ability of the DNA-particle-network to inhibit bacteria motility we collected bright-field microscopy videos and extracted the pixel-intensity standard deviation over 7 consecutive frames. This quantity is then time averaged over ~1500-frame (10 s) videos and normalized to extract a proxy for bacteria motility, we indicate as $\sigma$ (see definition in Methods). Colormaps in Fig. 5, showing the distribution of $\sigma$-values over a microscopy field of view, highlight a clear difference between samples with non-triggered and triggered particles after an 870-min incubation. If no trigger is added, $\sigma$ is larger and relatively uniform across the field of view, indicating sustained and unimpeded bacterial motion. In turn, when particle aggregation is triggered, $\sigma$-values are lower and non-uniform over the observed area, likely as a consequence of the hindering action of the sticky DNA web.

Figure 5d shows the time evolution of the frame-averaged $\sigma$ values ($\bar{\sigma}$) (see Supplementary Fig. 16 for the corresponding $\sigma$ colormaps). In both samples with and without trigger, $\bar{\sigma}$ exhibits an initial growth, which continues steadily throughout the experiment for non-triggered particles but soon reaches a plateau in the triggered sample, as network-forming particles capture the *E. coli*. The increase in the motility parameter $\bar{\sigma}$ with incubation time, also clearly visible in the $\sigma$ colormaps in Supplementary Fig. 16, is a consequence of bacterial growth, which can be also quantified by monitoring the average fluorescence emission from the mKate2 probe expressed by *E. coli*, as summarized in Fig. 5e. Here, triggered and non-triggered samples display a comparable and steady fluorescence increase, confirming growth and high-lighting that the clear difference in the motility parameter between the two samples (Figs 5c, d, and S16) is ascribable to bacteria immobilization, rather than a difference in growth rate. In fact, *E. coli* appear to use some element contained in the DNA particles, protected or unprotected, as a food source, as indicated by the absence of growth in bacteria samples lacking particles. This trend is confirmed by turbidity measurements shown in

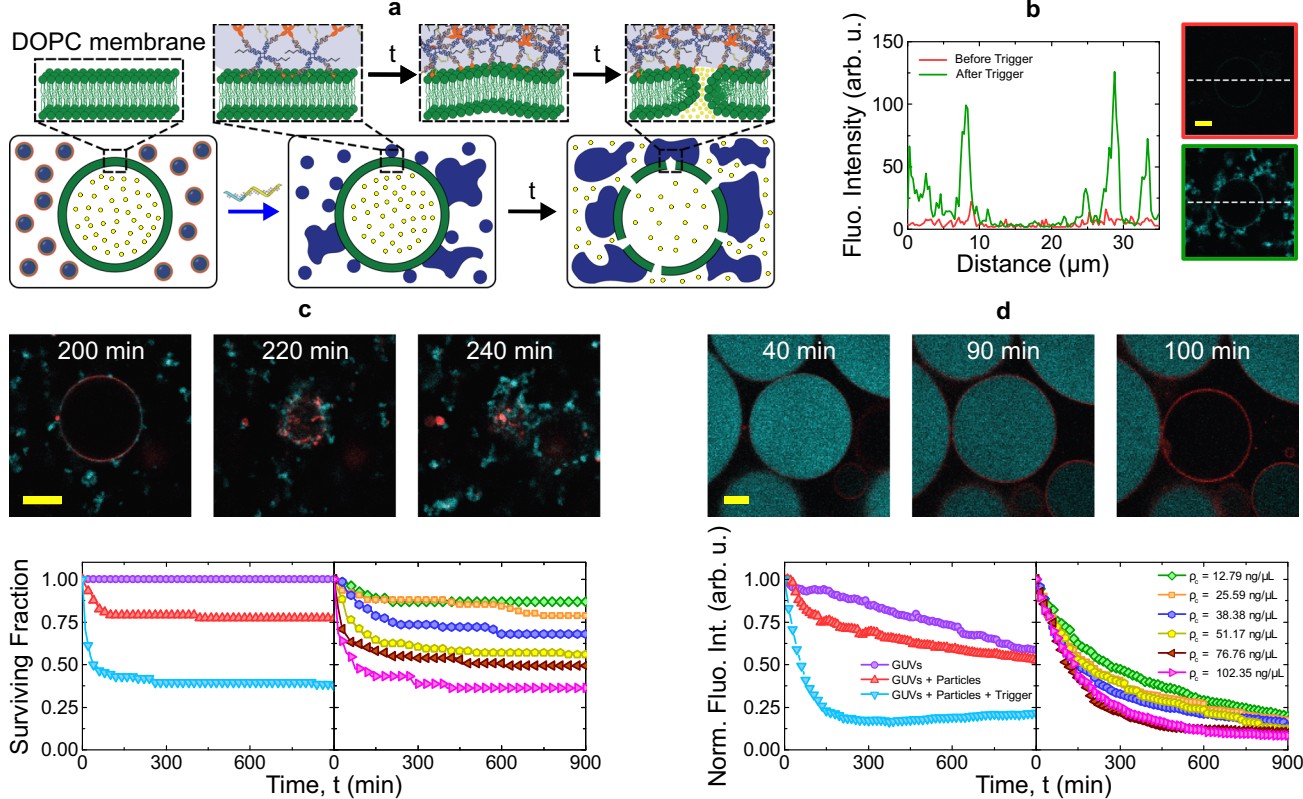

**Fig. 4 Unprotected particles trigger vesicle rupture and cargo release. a** Schematic representation of particle-induced disruption of Giant Unilamellar Vesicles (GUVs). Upon addition of the trigger strand and displacement of the hydrophilic corona, unprotected particles adhere to each other and to the GUVs owing to the hydrophobic nature of cholesterol molecules. DNA aggregates lead to GUV rupture and/or cargo release. **b** Confocal micrographs demonstrating the accumulation of amphiphilic DNA (fluorescein, cyan) onto a GUV before and after addition of the trigger strand. Scale bar 5 $\mu$m. Bottom: line profiles highlighting DNA accumulation onto the GUV surface. **c** Particle-induced GUV rupture as quantified from confocal time-lapse microscopy as the fraction of "surviving" GUVs over time. On the bottom, the left panel compares the effect of protected particles and unprotected ones exposed (at $t = 0$) to the trigger on otherwise stable GUVs. The right panel quantifies the effect of changing particle concentration $\rho_c$, expressed as the mass density of core C-stars. Legends as in panel **d** Top: confocal micrographs of membrane rupture induced by unprotected particles. Core C-stars are shown in cyan, the lipid membrane in red. Scale bar 10 $\mu$m. **d** Progressive leakage of fluorescein-sodium initially encapsulated in GUVs as quantified with confocal microscopy. Bottom: unprotected particles significantly increase the spontaneous leakage rate compared to control samples of unperturbed GUVs and those exposed to stabilized DNA particles. Top: Confocal micrographs demonstrating fluorescein-sodium (cyan) leakage following the adhesion of DNA particles (TXRED, red) onto GUVs. Scale bar 5 $\mu$m. Data shown in panels (**b**), (**c**), and (**d**) represent three and two independent experiments, respectively. The (mass) concentration of GUVs was $3.12 \pm 0.16$ g L$^{-1}$ for the data in panels (**c**, **d**).

Supplementary Fig. 17 and is aligned with previous reports about the ability of bacteria to draw energy from nucleic acids[45,46] and that of cholesterol functionalizations to enhance internalization[47–49]. Detailed inspection of the microscopy images, exemplified in Supplementary Fig. 18, confirms that bacteria can grow while embedded in the amphiphilic DNA network, and retain their structural integrity in this environment. The latter point indicates a greater robustness of the cell-wall against destabilization by the amphiphilic DNA network compared to bare lipid membranes, likely as a consequence of presence of the protective lipopolysaccharide layer[50].

## Discussion

In summary, we present the rational design of a multi-component system of amphiphilic and unmodified DNA nanostructures that self-assemble in one-pot reactions to form core–shell particles with programmable size. The stabilizing hydrophilic corona of the particles, instrumental in controlling their size via a two-step thermal annealing protocol, can be isothermally displaced by a DNA trigger. Corona removal leads to particle aggregation and, if lipid vesicles are present, to their disruption and enhanced permeability. Vesicle disruption occurs following the interaction of the bilayer with the

amphiphilic core of the unprotected DNA particles, and the formation of large DNA-lipid aggregates on the vesicles.

The responsive membrane-active DNA particles mimic the action of molecular and nanoscale biological agents evolved to damage cell membranes, and could find application as targeted cell-killing nanodevices with therapeutic value.

At late aggregation stages, the DNA particles form extended networks that envelop cell-sized vesicles, and are able to arrest the motion of swimming *E. coli*. The latter behavior is reminiscent of the ability of innate-immune cells to immobilize pathogens in sticky DNA webs[31–34], and could form the basis of antimicrobial technologies.

In addition, one could envisage the encapsulation of the particles as "synthetic organelles" in lipid-based artificial cellular system[51,52]. Here, the removal of the corona could lead to an apoptosis-like response, with the permeabilization of the synthetic-cell membrane and release of its content, a highly sought-after functionality for conditional drug release.

Finally, the approach introduced here for the production of size-controlled, stable, core–shell particles could be easily adapted to create smart drug delivery systems[53]. For this purpose, the blunt ends of the outer corona motifs could be labeled with cell-

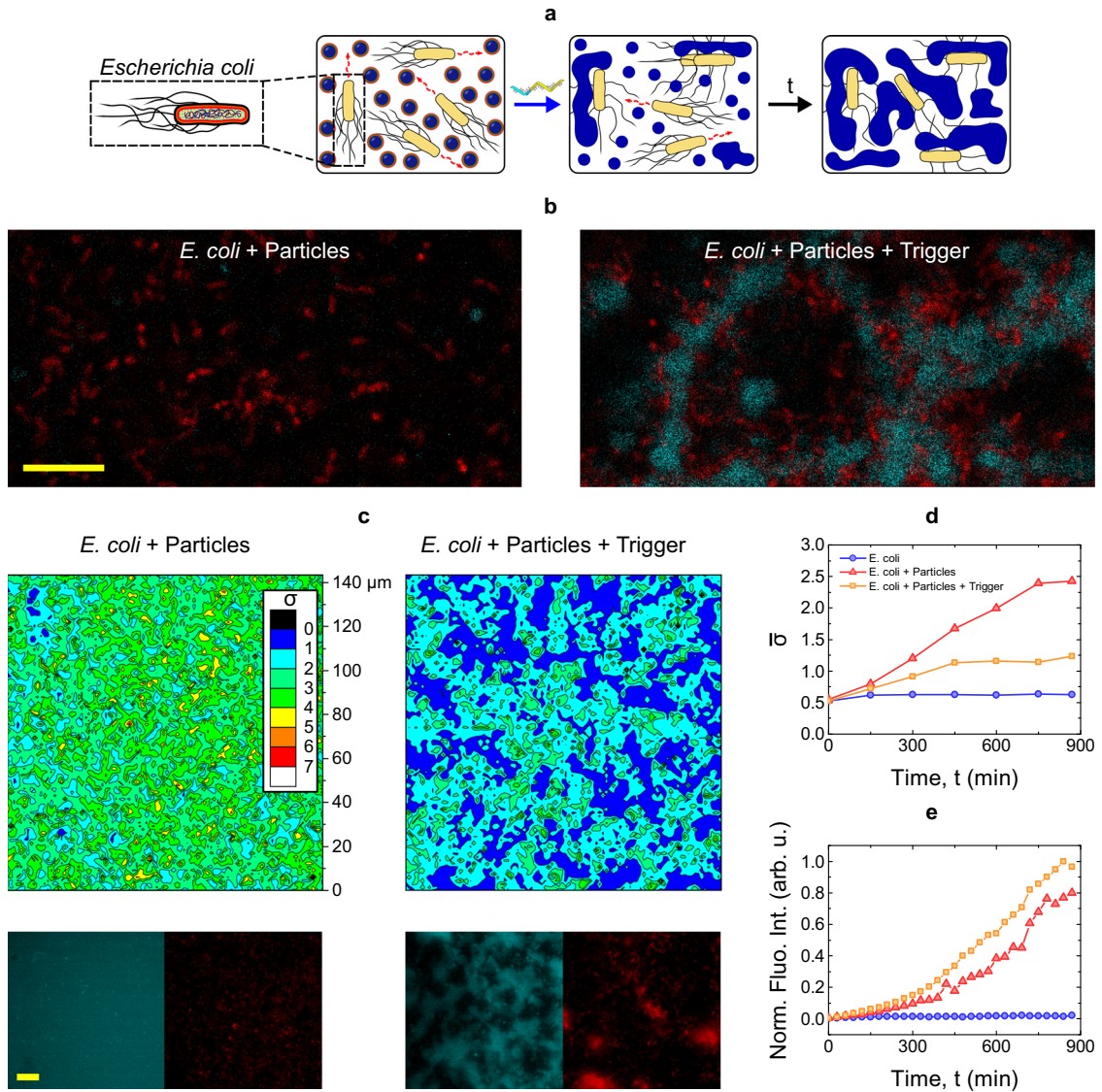

**Fig. 5 *E. coli* can be trapped by DNA networks formed by unprotected particles. a** Schematic representation of trigger-induced *E. coli* entrapment. Once activated by the addition of the trigger strand, particles assemble into a sticky DNA network. Swimming E. coli stick to, or become embedded in the aggregates, which renders them immobile. **b** Confocal micrographs demonstrating *E. coli* entrapment. Core C-stars (fluorescein) are shown in cyan, *E. coli* (mKate2) in red. Scale bar 10 $\mu m$. **c** Top: trigger-induced *E. coli* entrapment as quantified through a motility parameter $\sigma$ (extracted from microscopy videos, see Methods and main text for definition) for samples with and without the addition of trigger. The smaller and non-uniform $\sigma$-values detected in the presence of the trigger confirm the ability of the DNA aggregates to hinder *E. coli* motion. See Supplementary Fig. 16 for the $\sigma$-maps extracted at different incubation stages, where the absence of significant signal from the early-time maps confirms that Brownian motion from the particles has a comparatively negligible effect on $\sigma$. Bottom: epifluorescence micrographs in the DNA (cyan) and *E. coli* (red) channels collected in the corresponding fields of view. Scale bar 20 $\mu m$. **d** Time-trace of the frame averaged motility parameter $\bar{\sigma}$ for samples of *E. coli* and DNA particles (with and without trigger) and a control sample in which no DNA particles were present. The increase in $\bar{\sigma}$ observed in the presence of particles can be ascribed to bacterial growth, as confirmed in panel **e** and Supplementary Fig. 17. While $\bar{\sigma}$ continues to grow steadily in the sample with non-triggered particles, as more moving bacteria are generated, DNA-aggregation and *E. coli* entrapment cause the curve to plateau. **e** Normalized fluorescence intensity of *E. coli*-expressed mKate2 protein as extracted from epifluorescence images for the samples in panel (**d**). An increasing signal indicates bacterial growth, which occurs in the presence of DNA particles but is absent for the control DNA-only sample, suggesting that *E. coli* can use the DNA particles as a food source (see also Supplementary Fig. 17). Data shown in panels (**b**–**e**) was acquired in two independent experiments.

targeting peptides or aptamers[54,55], while the core motifs could host one or multiple therapeutic payloads either linked to the unused $\gamma$ overhang, partitioning in the hydrophobic cholesterol-rich pockets[35], or intercalating the DNA[56].

## Methods

**Design and handling of DNA oligonucleotides**. DNA nanostructures were designed using the NUPACK online tool[57]. All sequences are shown in

Supplementary Table 1. All DNA strands, apart from Core Strand 2 with an internal Texas Red DT modification (Eurogentec), were purchased from Integrated DNA Technologies (IDT). Cholesterol modified and fluorescently labeled oligo-nucleotides were purified by the supplier using high-performance liquid chroma-tography (HPLC), while the non-functionalized strands were purified using standard desalting. After delivery, the dehydrated DNA was reconstituted in phosphate-buffered saline (PBS, 137 mM NaCl, 2.7 mM KCl, 8 mM $Na_2HPO_4$, 2 mM $KH_2PO_4$, pH 7.4, Invitrogen Thermo Fisher Scientific) and syringe-filtered through polyethersulfone filters (0.22 $\mu m$ pore size, Millex) to remove large

particulate contaminants. Concentrations of all the reconstituted DNA strands were extracted from the absorbance at 260 nm as recorded with a Thermo Scientific Nanodrop 2000 UV–Vis spectrophotometer and using the extinction coefficients provided by the supplier. After preparation, all stock solutions were aliquoted into individual 50 μL volumes and stored at −20 °C until needed for sample preparation and further analysis.

**Preparation of non-cholesterolized nanostructures.** Samples for assessing the self-assembly and properties of individual nanostructures were prepared by replacing the cholesterol-modified strands with oligonucleotides of identical sequence, but lacking the cholesterol moiety.

All the required DNA strands were stoichiometrically mixed in 200 μL DNase free Eppendorf tubes. The concentration of each oligonucleotide are reported in Supplementary Table 2. Prepared mixtures were annealed in a Techne TC-512 thermal cycler by cooling down from 95 to 20 °C at a rate of −0.05 °C min⁻¹ to enable nanostructure assembly. Annealed samples were used within 24 h.

**Preparation of protected particles.** Oligonucleotides were mixed stoichiometrically in PBS with concentration of core strands equal to 1 μM (concentrations of all DNA strands are shown in Supplementary Table 2) and a total volume of 60 μL, in 500 μL DNase free Eppendorf tubes. In parallel, borosilicate glass capillaries (inner section of 4 mm × 0.4 mm, CM Scientific) were cleaned with a 2% Hellmanex III water solution (Hellma Analytics) and then washed twice with milli-Q water by sonicating for 15 min. After mixing, samples were loaded into the capillaries, which were capped on both sides with mineral oil and sealed by gluing them onto microscope coverslips (Menzel Gläser, 24 mm × 60 mm, No. 1) with 2-component fast-drying epoxy adhesive (Araldite).

For the preparation of macroscopic spherical aggregates (Fig. 2d), a sample was adhered to a custom-made Peltier-controlled copper plate, heated up to 90 °C and incubated at this temperature for 15 min. The temperature was then decreased to 85 °C at a rate of −1 °C s⁻¹, followed by a further cooling ramp from 85 °C to 60 °C at −0.017 °C min⁻¹. Afterwards, the sample was quickly cooled down to 35 °C, incubated for 3 h, and then brought down to 20 °C at a rate of −1 °C s⁻¹.

Samples for the preparation of large polyhedral (single-crystal) aggregates (Fig. 2e) were slowly annealed in a custom made, Peltier-controlled water bath by holding them at 95 °C for 30 min and them cooling down from 95 °C to 20 °C at a rate of −0.01 °C min⁻¹.

Protected particles of controlled size (Fig. 2a, b) were prepared on the Peltier-controlled copper plate by first incubating at 90 °C for 15 min, then quickly cooling down to 65 °C at −1 °C s⁻¹, incubating at this temperature for a variable growth time $t_g$ (depending on the sought particle size), and finally rapidly cooling down to 35 °C s⁻¹ to enable corona attachment (see Fig. 1). Particles were held at this temperature for 15 min and then further cooled down to 20 °C. For experiments aimed at monitoring particle-size dependence on $t_g$ and their long-term stability (Fig. 2a, b) the whole two-step self-assembly protocol described above was performed on a tailor-made Peliter microscope stage mounted onto a Nikon Eclipse Ti-E inverted microscope.

Note that changing oligonucleotide concentrations with respect to the values reported in SI Table 2 may lead to reduced particle stability. In particular, we have observed that stability against aggregation is compromised if the concentration ratio between core and corona strands is increased by a factor 3.5 or more with respect to the reported protocol.

**Agarose gel electrophoresis.** AGE was used to verify the correct assembly of DNA motifs, using their non-cholesterolized versions (see Supplementary Fig. 1).

After nanostructure annealing, samples were diluted in TBE buffer (pH 8.3, 89 mM Tris-borate, 2 mM EDTA, Sigma-Aldrich/Merck) and mixed with a loading dye (Thermo Fisher Scientific) at 5:1 ratio, resulting in the final DNA concentration of 75 ng μL⁻¹.

Gels were prepared at 1.5 wt% agarose (Sigma-Aldrich) in TBE, and pre-stained with SYBR safe DNA gel stain (Invitrogen/Thermo Fisher Scientific). Wells were loaded with 10 μL of diluted sample, corresponding to 750 ng of DNA. Two wells were dedicated to a 100 bp DNA reference ladder (Thermo Fisher Scientific). Electrophoresis was carried out at 75 V, equivalent to 3 V cm⁻¹, for 120 min. Afterwards, gels were imaged with a GelDoc-It imaging system, using the Vision Works software.

The analysis of acquired images was performed using a tailor-made Matlab script.

**UV melting curves of non-cholesterolized motifs.** UV–Vis spectrophotometry, performed on a temperature-controlled Varian Cary 50, was used to determine the melting temperature of DNA nanostructures (Supplementary Fig. 3), using their non-cholesterolized versions.

Samples were prepared according to the procedure described above (see also Supplementary Table 2), but omitting the annealing step, and then diluted to limit absorbance at 260 nm to below 1. Quartz cuvettes were filled with 1.2 mL of sample, capped with 300 μL of mineral oil and sealed with a Parafilm-wrapped PTFE stopper to prevent evaporation.

Absorbance at 260 nm was measured using the Cary WinUV Thermal Application software over two contiguous temperature ramps: first cooling from 95 to 20 °C and then heating back up to the same initial temperature. Heating/Cooling rates were set to ±0.02 °C min⁻¹.

For each of the ramps, the melting temperature ($T_M$) was determined by fitting the lower and upper plateaus with a straight line and then calculating the intersection of the median between these fits and the experimental data[58], using a tailor-made Matlab script.

**DLS of individual DNA motifs and particles.** DLS was used to confirm the successful folding of non-functionalized DNA constructs and to acquire the size distribution of protected particles (Figs. S2, S6).

DLS measurements were performed with a Malvern Zetasizer Nano ZSP analyzer (scattering angle fixed at 173°, 633 nm He–Ne laser), using the Malvern Zetasizer software. For these experiments, 100 μL of the samples, either non-cholesterolized motifs or particles, were loaded into an ultra-low volume quartz cuvette (ZEN2112, Malvern) and then three measurements, each consisting of twelve data runs, were taken at room temperature.

**TEM of protected particles.** TEM micrographs of protected particles of different sizes, featured in Figs. 2c and S9, were obtained using a JEOL JEM-2100F transmission electron microscope, fitted with a Gatan Prius SC 1000 camera (2 × 4k), and the Gatan Digital Micrograph software. An aliquot (5 μL) of each sample was placed onto a 45 s glow-discharged 200 square mesh copper grid (Agar Scientific). The grids were blotted with filter paper after 1 min deposition and imaged directly without the use of any staining agent to avoid sample instabilities observed with the use of a 2% w/v uranyl acetate solution. The samples were imaged by using a low electron dose rates to avoid sample damage. All the registered TEM micrographs were analyzed in ImageJ.

**Confocal Microscopy of large particles.** Confocal micrographs of spherical and single-crystal large aggregates with a visible core–shell structure (Fig. 2d, e) were recorded using a Leica TCS SP5 laser scanning confocal microscope equipped with a HC PL APO CORR CS 40 × /0.85 dry objective (Leica), and the Leica Application Suite Advanced Fluorescence software. Prior to imaging, samples (60 μL) were removed from the capillaries into which they had been annealed and washed twice to remove excess corona motifs (see also Supplementary Fig. 4). Washing was carried out by 2 × dilution in PBS buffer followed by centrifugation (30 min at 420 g) and subsequent supernatant replacement with fresh PBS buffer (90 μL). Small volumes of washed samples (20 μL) were loaded into silicon incubation chambers (6.5 mm × 6.5 mm × 3.5 mm, Grace Biolabs FlexWells) and sealed with DNase free tape (Grace Biolabs FlexWell SealStrips) to prevent evaporation. Signals from the fluorescein-labeled core motifs and outer corona motifs tagged with Alexa Fluor 647 were recorded by exciting with an Ar-ion laser line at 488 nm and a HeNe line at 633 nm.

**Bright field and epifluorescence imaging of protected particles.** Bright field microscopy images (Figs. 2a, 3b) and videos of DNA particles were acquired with fully motorized and programmable Nikon Eclipse Ti-E inverted epifluorescence microscope, equipped with Perfect Focusing System (Nikon) and Grasshopper3 GS3-U3-23S6M camera (Point Gray Research), using a CFI Plan Apochromat λ 40 × /0.95 NA dry objective (Nikon). Temperature was controlled with the aforementioned copper-plate Peltier stage, and transmission imaging enabled by a small aperture. A transparent sapphire window (with large thermal conductance) was sandwiched between the sample and the copper plate to ensure a uniform temperature across the sample. Epifluorescence images of fluorescein-labeled particles (Fig. 3b) were recorded with the same setup using blue LED illumination and a GFP filter cube (Semrock).

**Assessment of size and stability of DNA particles with DDM.** DDM was used to assess the size and stability of DNA particles formed with various growth times ($t_g$)[38,39]. Samples were processed while on the microscope with the thermal annealing process described above, and high frame-rate videos (150 fps, 7 s) were recorded after the final annealing stage ($T = 20$ °C) to be used for DDM analysis of particle hydrodynamic size. Videos were recorded at regular intervals for up to 1 h (Fig. 2b) or 50 h (Supplementary Fig. 7) to assess particle stability. For the sample incubated at $T = 65$ °C for $t_g = 1800$ s (Supplementary Fig. 7) two further measurements were taken after 8 and 14 days to confirm long-term stability.

Data analysis was performed using a tailor-made Matlab script.

Briefly, let us indicate as $I(\mathbf{r}, t_0)$ a microscopy frame collected at time $t_0$, and with $\Delta I(\mathbf{r}, t_0, \tau) = I(\mathbf{r}, t_0 + \tau) - I(\mathbf{r}, t_0)$ the difference between two such frames separated by a time interval $\tau$. It can be demonstrated that, for Brownian particles, the time-averaged and azimuthally-averaged Fourier transform of $\Delta I(\mathbf{r}, t_0, \tau)$ obeys[38]

$$\Delta I(q, \tau) = A(q)\left[1 - \exp\left(-D\, q^2\, \tau\right)\right] + B(q), \qquad (1)$$

where $D$ is the Brownian diffusion coefficient, $q$ the module of the 2D spatial wave vector, and $A(q)$ and $B(q)$ are functions dependent on the static scattering

properties of the particles, the instrument optics, and camera noise. In DDM, the diffusion coefficient is extracted by fitting experimental data for $\Delta I(q,\tau)$ to Eq. (1) over a suitable $q$-range. The hydrodynamic radius can then be extracted from $D$ using the Stokes-Einstein equation. Note that for non-Brownian samples, e.g. when large gel-like particle aggregates emerge, a deviation from the $\propto q^2$ functional form is expected in the argument of the exponential in Eq. (1)[59]. In our experiment, this occurs when tracking triggered particle aggregation (Fig. 3b). However, for consistency, we still fit $\Delta I(q,\tau)$ with the Brownian functional form, aware that the extracted hydrodynamic radius represents an effective value with little physical meaning. Still, this effective hydrodynamic size is a good indicator for the occurrence of particle aggregation.

As expanded in Supplementary Discussion 2, Eq. (1) can be modified with a truncated cumulant expansion to extract sample polydispersity. Such analysis is applied to the DDM data for particles formed at relatively long growth times, as summarized in Supplementary Fig. 5.

**AGE and DLS assessment of the trigger detachment of core, and corona motifs**. Two separate experiments were carried out with AGE and DLS to test the toeholding mechanism detaching core from corona motifs (see Figs. S10, S11).

Electrophoresis was carried out using a previously described procedure and AGE setup. In this experiment, annealed samples were additionally mixed with the trigger strand and left for 5 h before adding the loading dye and injecting them in the wells.

The obtained images were analyzed with a tailor-made Matlab script.

DLS measurements were performed with the aforementioned DLS setup. For these experiments, 100 $\mu$L of the samples were loaded into an ultra-low volume quartz cuvette and then the trigger was added at 5 or 10 × excess compared to the inner corona motif. In all cases, the final volume was adjusted to obtain a core motif concentration of 0.85 $\mu$M. The average hydrodynamic diameter detected was then monitored over time.

**Triggered corona displacement and particle aggregation**. For experiments on trigger-induced shell release, followed by particle aggregation, two samples prepared with $t_g = 300$ s were extracted from the capillaries where annealing took place and placed in silicon incubation chambers (6.5 mm × 6.5 mm × 3.5 mm, Grace Biolabs Flexwells). The trigger strand was added to one of the samples at 10× excess compared to the inner corona motif, before sealing the chambers (Grace Biolabs Flex Well Seal Strips) and imaging for 700 min. The remaining particle sample was used as a control. Bright field videos (150 fps, 7 s) were collected every 30 min and used for DDM analysis of the apparent hydrodynamic radius (Fig. 3b). Epifluorescence Z-stacks were recorded every 10 min and used for the aggregation analysis in Fig. 3b.

**GUV preparation**. GUVs with and without cargo molecules were prepared from 1,2-dioleoyl-sn-glycero-3-phosphocholine lipids (DOPC, Avanti Polar Lipids) using a modified protocol for the emulsion-transfer method[60–62].

Two 1.5 mL glass vials were cleaned by sonicating three times for 15 min each, once in 2% Hellmanex III water solution and then twice in milli-Q water. A small volume (40 $\mu$L) of 25 mg mL$^{-1}$ lipid solution in chloroform was added to 500 $\mu$L of paraffin oil (Sigma-Aldrich), previously pipetted into the clean vial, and vortexed for 2 min. For the preparation of fluorescently tagged GUVs, a 10 mg mL$^{-1}$ mixture of DOPC lipids in chloroform with 0.8% (molar ratio) Texas Red 1,2-Dihexadecanoyl-sn-Glycero-3-Phosphoethanolamine (Texas Red - DHPE, Invitrogen) was used instead. In the next step, the solution was incubated for 1h at 85°C to induce chloroform evaporation, and let to cool down to room temperature. The overall lipid concentration in the oil solution was 2 mg mL$^{-1}$ for not tagged and 0.8 mg mL$^{-1}$ for fluorescently tagged GUVs. Afterwards, 250 $\mu$L of the lipid-oil solution was transferred to a previously cleaned glass vial and mixed with 25 $\mu$L of the Inside-solution (I-solution, 300 mM sucrose in milli-Q water and 1 mM fluorescein-sodium with 299 mM sucrose in milli-Q water for GUVs without and with encapsulated cargo, respectively) by vortexing for 1-2 min to generate white and turbid emulsion. The resultant mixture was layered on top of 150 $\mu$L of the Outside-solution (O-solution, 300 mM glucose in milli-Q water) in 1.5 mL DNase free Eppendorf tubes, and centrifuged at 9000 $g$ for 30 min. The osmolarities of the I-solution and O-solution were chosen to match that of the PBS buffer used in the production of protected particles. After centrifugation, the supernatant was removed from the sample, leaving 50 $\mu$L of the final solution, which was subsequently diluted by adding 100 $\mu$L of the O-solution. Special attention was given to remove all of the paraffin oil from the sample before diluting it. Prepared vesicles were stored at room temperature and used within a day.

**Triggered GUV disruption and cargo release**. For experiments on trigger-induced lipid membrane rupture and fluorescein-sodium release (Fig. 4c, d, left), three incubation chambers previously passivated with BSA (Bovine Serum Albumine, Sigma) were filled with 30 $\mu$L of previously prepared GUVs. Fluorescently tagged (process visualization) and non-tagged cargo-loaded (membrane rupture efficiency quantification) GUVs were used for the membrane-disruption assay, while non-fluorescent GUVs loaded with cargo molecules were used for the leakage assay. 90 $\mu$L of the PBS solution containing protected particles, prepared with $t_g = 15$ s, were then pipetted into two wells, while the third was filled with an

identical volume of particle-free PBS buffer, and used as a control. A small volume of a concentrated PBS solution of the trigger strand was added to one of the particle-containing wells resulting in a 10× excess compared to the concentration of the inner corona motif. The volume of the remaining two samples, including the control GUV-only sample, was adjusted to the same value as the trigger-containing chamber by adding fresh PBS. The chambers were then sealed with DNase-free tape to prevent evaporation.

For experiments intended to unravel the influence of particle size on the membrane rupture efficiency (see Supplementary Fig. 13), the aforementioned sample preparation protocol was followed, changing only the particle size by extending $t_g$ to 180 s and 600 s.

For experiments aimed at investigating the effect of particle concentration (Fig. 4c, d, right) six wells were filled with GUV solution as described above, before adding different volumes of the particle-containing and trigger solution to achieve the sought particle concentrations. The overall volumes were adjusted to the same value by adding fresh PBS.

Confocal Z-stacks were recorded every 10 min (Fig. 4c, d, left panels; Supplementary Fig. 12), 15 min (Fig. 4c, d, right panels) and 16 min (Supplementary Fig. 13). For visualization of the particle-induced membrane rupture process (Figs. 4c and S12), signals from the fluorescein-labeled core motifs and membranes tagged with DHPE Texas Red were recorded by exciting with an Ar-ion laser line at 488 nm and a HeNe line at 594 nm. For the fluorescein-sodium leakage assay (Fig. 4d) and the particle-induced GUV rupture assay (Figs. 4c, S13), fluorescent cargo molecules and Texas Red-tagged core motifs were excited using the 488 nm and 594 nm lines, respectively.

All the above-mentioned data were analyzed using ImageJ and Matlab.

**LUV preparation**. LUVs with encapsulated calcein were prepared from DOPC lipids using the extrusion method, before removing unencapsulated calcein with size exclusion chromatography.

Glass vials (1.5 and 15 mL) were cleaned with the 3-step sonication protocol described above. A clean 1.5 mL vial was filled with 200 $\mu$L of a DOPC solution in chloroform (25 mg mL$^{-1}$), and placed in a vacuum desiccator for 1 h to induce chloroform evaporation. Afterwards, the dry lipid film was reconstituted in 400 $\mu$L of a solution containing 80 mM calcein and 220 mM sucrose in milli-Q water, and gently vortexed for 10 min at 400 rpm. The sample was then subject to five freeze-thaw cycles using liquid nitrogen and a heating block set to 80 °C. Finally, the mixture was extruded 25 times through Whatman filters (10 $\mu$m pore size, Fisher Scientific) and membranes (1 $\mu$m pore size, Fisher Scientific) using an mini-extrusion kit from Avanti Polar Lipids.

In order to remove non-encapsulated calcein, the sample was passed through a size exclusion column (Pierce Disposable Columns, 5 mL, Thermo Fisher Scientific) filled with a Sephadex G50 gel filtration medium (Sigma-Aldrich) and a 300 mM glucose solution in milli-Q water. Collected vesicles were stored at room temperature and used within a day.

**Calcein leakage assay from LUVs**. Fluorescence measurements for the calcein leakage assay from LUVs (Supplementary Fig. 15) were carried out with a FLUOstar Omega plate reader, using the Omega Control microplate reader data collection/analysis software. LUVs with encapsulated calcein were placed in three sterile, DNase free 96 Well Cell Culture Microplates (CELLSTAR Clear 96 Well Cell Culture Microplates, Greiner Bio-One), which were passivated with BSA beforehand. Subsequently, protected particles prepared with $t_g = 15$ s were pipetted into two of these wells. The remaining well was filled with a matching amount of PBS buffer to account for any changes in osmolarity or ionic strength caused by the particle addition. Afterwards, an aliquot of trigger strand solution in PBS was added to one of the particle-containing wells (10 × excess compared to the inner corona motif), and the microplate was covered with a lid provided by the manufacturer to prevent evaporation. The fluorescence signal was measured every 30 s for 4 h. Excitation and emission wavelengths were set using Green Fluorescent Protein excitation (485 nm) and emission (520 nm) filters.

**Quantification of lipid concentration in GUVs and LUVs**. A phosphatidylcholine (PC) enzymatic assay was used to quantify lipid concentration in GUV samples prepared via the emulsion-transfer method and LUV samples prepared using the extrusion method. A PC enzymatic assay kit (Sigma-Aldrich) was used following the manufacturer's instructions. GUVs and LUVs were diluted (1:50) in a reaction master mix, which contained PC hydrolysis enzyme, a development mix, assay buffer and fluorescent peroxidase substrate. Samples were incubated for approximately 30 min, and then absorbance at 570 nm ($A_{570}$) was measured for an hour with a FLUOstar Omega plate reader. $A_{570}$ values were subsequently averaged and used to determine the concentration of PC in GUVs/LUVs using a calibration curve acquired from standard samples of PC as instructed by the manufacturer. All samples were blanked against absorbance measurements of GUVs/LUVs diluted in the reaction mix but in the absence of the enzyme.

The assay produced a lipid (PC) concentration of 3.12 ± 0.16 g L$^{-1}$, 1.66 ± 0.14 g L$^{-1}$ and 0.96 ± 0.12 g L$^{-1}$ for non-tagged and fluorescently tagged GUVs, and LUVs, respectively. The calculated (mass) concentration of non-tagged GUVs was

used to express the "lethality" of particles as a function of C-Star/lipid molar ratio (see Fig. S14).

**Bacterial strains and growth conditions.** Bacteria cells (*Escherichia coli*, *E. coli*) MG1655 strain, which is rendered highly motile by the presence of an insertion sequence element in the flhD operon[63] used in the entrapment assay were grown in M63 medium (13.6 g L$^{-1}$ KH$_2$PO$_4$, 0.5 mg L$^{-1}$ FeSO$_4$ 7H$_2$O, 0.5 g L$^{-1}$ MgSO$_4$ 7H$_2$O, 1.27 mg L$^{-1}$ Thiamine, 2.64 g L$^{-1}$ (NH$_4$)$_2$SO$_4$ and 0.5% w/v Glucose) at 37 °C with shaking at 220 rpm and in balanced growth, as this ensures reproducibility of the motile cells fraction. Balanced growth was obtained by first growing single colonies in LB medium (10 g L$^{-1}$ Tryptone, 5 g L$^{-1}$ Yeast extract, 0.5 g L$^{-1}$ NaCl) for 3-4 h. Cells were then transferred to M63 medium at a final dilution of 10$^{-7}$ and grown to an OD of 0.2–0.3. Before the experiments, bacteria were pelleted via centrifugation at 8000 *g* for 2 min, washed twice in Berg's Motility Buffer (BMB) (6.1 mM Na$_2$HPO$_4$, 3.9 mM NaH$_2$PO$_4$, 0.01 mM EDTA, 0.5% w/v Glucose) and diluted to a final OD of 0.15. The bacteria carried the pTP20-mKate2 plasmid[64], which encodes for the mKate2 fluorescent protein marker that is constitutively expressed in the cytoplasm.

**Bacteria entrapment assay.** To test whether the DNA aggregates could entrap bacteria (Figs. 5b, c, and S16), three BSA-passivated wells were filled with 60 µL of the previously prepared *E. coli* (MG1655 strain) in BMB solution (OD = 0.15). Two of the chambers were then topped with 60 µL of the PBS solution containing protected particles ($t_g$ = 15 s). The third well, used as a control sample, was filled with an identical volume of PBS buffer. Afterwards, a small volume of the trigger strand in PBS solution was added to one of the *E. coli* containing chambers to initiate aggregation. The trigger strand was in a 10× excess compared to the concentration of the inner-corona motif. The volume of the two remaining samples was adjusted with fresh PBS buffer to the same value as the trigger-containing sample. The wells were then sealed with DNase-free tape to avoid evaporation.

All samples were imaged with the aforementioned bright field and epifluorescence microscopy setup by acquiring a set of high frame-rate videos (150 fps; 10 s) and epifluorescence micrographs at regular intervals of 30 min for 870 min (Figs. 5c and S16). Signals from the fluorescein-labeled core motifs and mKate2 expressed by *E. coli* were recorded using blue LED illumination/GFP filter (Semrock) and green LED/Texas Red filter (Semrock), respectively. After the experiment, all samples were additionally imaged using the previously mentioned confocal microscope (Fig. 5b). The Ar-ion laser line at 488 nm (particles) and the HeNe line at 594 nm (*E. coli*) were used for excitation.

The quantity σ used as a proxy for bacteria motility in Figs. 5c and S16 (colormaps) was computed with a tailor-made Matlab script from bright field videos as $\sigma = \langle \sqrt{\frac{\sum_{m=-3}^{3}[I(n+m)-\bar{I}]^2}{7}} \rangle_n \frac{\sigma_{max}}{7}$, where $I(n)$ is the *n*th frame of a bright field video, $\bar{I} = \frac{\sum_{m=-3}^{3} I(n+m)}{7}$, $\langle \dots \rangle_n$ indicates a rolling average over the entire video, and $\sigma_{max}$ represents the highest intensity standard deviation value obtained in all the videos. The frame-averaged parameter $\bar{\sigma}$ (Fig. 5d) was computed by averaging over all the pixels in the frame.

**Bacterial growth in the presence of DNA.** Bacterial growth in various conditions (Supplementary Fig. 17) was measured by monitoring optical density with a FLUOstar Omega plate reader, using the Omega Control microplate reader data collection/analysis software. Cells were initially grown and washed as described above, then resuspended in BMB and inoculated with various DNA solutions in PBS in a DNase free, flat-bottom 96-well plate (CELLSTAR Black 96 Well Cell Culture Microplates, Greiner Bio-One) at a final OD of 0.15. Optical density was automatically measured in the plate reader for 1090 min at 7 min intervals at 37 °C, with shaking at 700 rpm in double-orbital mode.

**Reporting summary.** Further information on research design is available in the Nature Research Reporting Summary linked to this article.

## Data availability

A fully representative selection of the data underlying these findings can be accessed free of charge at https://doi.org/10.17863/CAM.70338. A detailed description of said data is provided in the repository. Owing to the very large data volume associated with the article, particularly high-resolution microscopy images and high frame-rate videos, it would have been unpractical to upload the complete datasets. These are however available upon request to the corresponding author.

## Code availability

The Matlab and Python scripts used to analyze the agarose gel images, melting curves, DDM data, confocal micrographs of GUV samples and bright-field videos of *E. coli* samples are available from the corresponding author upon reasonable request.

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

## Acknowledgements

M.W. acknowledges support from the Engineering and Physical Sciences Research Council (EPSRC), and the Department of Physics at the University of Cambridge (the McLatchie Trust fund). L.D.M. acknowledges support from a Royal Society University Research Fellowship (UF160152) and from the European Research Council (ERC) under the Horizon 2020 Research and Innovation Programme (ERC-STG No 851667 NANOCELL). L.M. and P.C. acknowledge funding from the EPSRC (EP/T002778). C.C. acknowledges support from the Wellcome Trust Institutional Strategic Supporting Fund for an ISSF Springboard Fellowship (RSRO_67869). R.R.-S. acknowledges support from the Mexican National Council for Science and Technology (CONACYT, Grant No. 472427), the EPSRC CDT in Nanoscience and Nanotechnology (NanoDTC, Grant No. EP/L015978), and from the Cambridge Trust. The authors thank Nicola Pellicciotta and Luigi Feriani for providing the DDM Matlab script.

## Author contributions

M.W. and R.A.B. designed the DNA sequences and developed the experimental protocols. M.W. conducted all the experiments apart from TEM (C.C.) and DLS of particles (W.T.K.), supported by R.A.B., L.M., and R.R.-S. M.W. analyzed the data aided by W.T.K. M.W. and L.D.M. wrote the paper. L.D.M. designed and supervised the research aided by P.C. and R.A.B. All authors discussed the results and edited the paper.

## Competing interests

The authors declare no competing interests.
