## [Peer Review File · Nature Communications]

REVIEWER COMMENTS

Reviewer #1 (Remarks to the Author):

This manuscript by Michal Walczak et al describes an approach to prepare DNA-based particles capable of inducing liposomes rupture in a controllable fashion. The particles are composed of a core, which is prepared following a protocol well established by authors in their previous papers, and a stimulus-responsive shell which stabilises the particles and prevents their interaction with lipid vesicles. Shell-free particles interact with lipid vesicles due to the presence of cholesterol tags, causing their disruption or producing their entrapment within a gel-like network. Corona removal can indeed be controlled by a toehold-mediated strand displacement process which is of relevance to enable the controlled subsequent disruption of liposomes. Authors envision these particles could act as mimetic systems of biological entities capable of triggering cell membrane disruption as well as smart drug delivery systems. The originality of the described approach, their potential applications in biomedical science and the quality of the paper make this manuscript suitable for the scope of Nature Communications. Results, discussion, conclusions and methods are overall presented in a clear fashion. References are precise and concise. The following aspects need to be addressed by authors to enhance the quality of the manuscript before publication.

1. Authors introduce a γ motif at the core structure indicating that this can be used for loading molecular cargoes. To enhance the utility of the presented particles, authors should include an experimental example of this cargo to demonstrate such a functionality. The introduction of cargoes may modify the structural features of the DNA particles. Can the authors comment on this?

2. Fig S2 includes DLS values in number. Authors should also provide the values obtained in intensity.

3. Authors prepare these particles utilising DNA oligonucleotides at a micromolar concentration. Have authors explored the concentration range at which these particles are formed? Is it possible to scale up their production at larger volumes? The size of the DNA particles reaches a plateau after 50 hours at room temperature. Can authors comment on their colloidal stability on the order of days? How crucial is temperature in maintaining stability? All these features are important aspects which should be discussed specially considering potential future applications that according to authors these materials can bring as smart drug delivery systems.

4. Large particles cannot be observed by TEM and authors indicate this may be due to TEM sample preparation. Authors should consider alternative ways to image them such as AFM.

5. In figure 4c and d, the effect of DNA particle concentration on GUVs disruption and calcein leakage is shown. Fig S12 shows this DNA particle concentration follows Hill function. The molar concentration of GUVs has also a direct effect on these dose-dependent "lethality" but it is not included in the text. Likewise, LUVs concentration is not shown. Authors should include these data.

Reviewer #2 (Remarks to the Author):

In this manuscript, DNA hydrogel particles are developed that contain a hydrophobic cholesterol functionality. These cholesterol groups are initially protected at the surface of the particle by capping DNA structures that can be removed by toe-hold strand displacement by an additional DNA "trigger". Once deprotected, these hydrophobic hydrogel particles further aggregate into larger gel structures and/or interact with coexisting lipid vesicles in their surroundings. This destabilises the GUVs causing them to leak and collapse - a common effect when GUVs are subjected to a high concentration of membrane-active substances. Some GUVs also become encased in a complete gel corona such that they arrest in state. None of these results are particularly unexpected or surprising.

The paper is clearly written and the experiments appear to be well performed, analysed and presented. The effect of these hydrogel particles on GUVs is clear. What I do struggle with is the "why?". This is something the authors should aim to expand on and clarify for the paper to be of broad interest. At present I cannot put my finger on either the new fundamental scientific understanding that is gained or the exciting new application that this work enables. It currently falls a little into the no-man's-land between the two, achieving neither. I find the analogies with natural biological processes tenuous and the work does not add any new insight to these. Similarly the potential applications in biosensors, therapeutic technologies or synthetic biology are highly speculative and none are well borne out such as to become convincing as real or viable. Perhaps focus on one of these, fleshing out the details and a proof of concept demonstration would significantly strengthen the interest of the paper for a broad multidisciplinary audience.

Reviewer #3 (Remarks to the Author):

Recommendation: Publication premature at current form

Comments:

Walczak et al. assembled DNA-based particles with amphiphilic core and hydrophilic shell from short single-stranded DNA molecules. The particles were assembled in one-pot reaction with three temperature steps. By increasing the incubation time at 65°C, particles with bigger sizes were assembled. The particles were characterized with differential dynamic microscopy and fluorescence microscopy. Notably, the hydrophilic shell can be displaced by a strand-displacing reaction, leading to destabilization of the particles. These deprotected particles showed enhanced rupturing activity of GUVs.

Although the one-pot assembly of core-shelled particle out of oligonucleotides is impressive, I do not feel the claims in the manuscript are justified by the data presented in the manuscript. Especially, high amount of leaks in the figure 4b concerns me. Also, the protected particles grow significantly in terms of their volume over time (Figure S7) and shows significant activity for rupturing GUVs (Figure 4a). Therefore, I believe the stability and activity of the DNA particles should be improved before it can be published in Nature Communications. I have put more detailed moments below.

In figure 2a, particles are difficult to see. Please consider enlarging the snapshot and reducing the field of view.

I do not completely agree with the stability argument presented in figure 2b. The particles grow 10~20% in terms of hydrodynamic radius within one minute. Compared to the growth curve presented in Figure 2a, it is still significant and fast. There are some long-term stability curves in Figure S7, but it should be investigated further and incorporated in the protocol for figure 4. The particles might behave better if they are incubated for ~400 min before rupturing experiments.

Also, considering the hydrodynamic radius $r_h \sim V^{1/3}$, the change in volume could be significant over time.

It is difficult to see and therefore claim a monotonic increase in Figure S8. If the authors still want to use TEM, please consider CryoTEM. But I do not feel a TEM is necessary.

I am not sure the fluorescence intensity on surface could serve as a quantitative marker for aggregation in Figure 3b. It clearly shows kinetics involving aggregation and sinking of the particles which is difficult to correlate directly with growth. I suggest either use mean fluorescence intensity of particles or develop a mathematical model for the data.

Figure parts labelling should be a, b rather than a. b.

The labelling scheme of lines in Figure 4c and d could be improved. Using the same colour and shape combination to mark different conditions and concentrations at the same time could confuse readers.

The Figure 4 should be the highlight of the manuscript but it is bit worrying. First of all, calcein should not leak from GUV in a normal condition. In the Figure 4b, the GUV without any trigger releases calcein at 100 minutes timescale. However, the preparation of the GUV and chamber should take hours. Then why the GUVs still have calcein inside after hours of preparation? This suggest the GUVs in the microfluidic chamber is somehow destabilized. The destabilization could come from difference in ionic strength or high density of biotin/streptavidin on the membrane/surface. Also, if the authors find it difficult to prevent leakage of calcein, other dyes such as sulforhodamine B can be used as well.

Is there any reading to use the particles grown for 15 seconds to rupture the membrane? Have authors tried particles with bigger sizes to rupture the membrane?

Author's Response to the Reviewers

Reviewer 1

This manuscript by Michal Walczak et al describes an approach to prepare DNA-based particles capable of inducing liposomes rupture in a controllable fashion. The particles are composed of a core, which is prepared following a protocol well established by authors in their previous papers, and a stimulus-responsive shell which stabilises the particles and prevents their interaction with lipid vesicles. Shell-free particles interact with lipid vesicles due to the presence of cholesterol tags, causing their disruption or producing their entrapment within a gel-like network. Corona removal can indeed be controlled by a toehold-mediated strand displacement process which is of relevance to enable the controlled subsequent disruption of liposomes. Authors envision these particles could act as mimetic systems of biological entities capable of triggering cell membrane disruption as well as smart drug delivery systems. The originality of the described approach, their potential applications in biomedical science and the quality of the paper make this manuscript suitable for the scope of Nature Communications. Results, discussion, conclusions and methods are overall presented in a clear fashion. References are precise and concise.

Response: We thank the reviewer for the positive comments on our submission.

The following aspects need to be addressed by authors to enhance the quality of the manuscript before publication.

Reviewer Point 1.1 — Authors introduce a γ motif at the core structure indicating that this can be used for loading molecular cargoes. To enhance the utility of the presented particles, authors

should include an experimental example of this cargo to demonstrate such a functionality. The introduction of cargoes may modify the structural features of the DNA particles. Can the authors comment on this?

Response: In the revised version of the manuscript we included an entire new section and Figure (5) on the application of network-forming DNA-particles for arresting the motion of bacteria. While this functionality does not make direct use of the domain 'y, we would argue that it adds substantial value to our contribution and proves the potential usefulness of our system. We prefer thus to reserve the experiments on embedding functional cargoes for a future publication, so to avoid excessively diluting the message of the present submission. However, we can still address the Reviewer's question on the structural impact of anchoring cargoes within the amphiphilic DNA frameworks. Indeed, in another recent publication exploring C-star self-assembly in macroscopic single crystals [Brady R. A. *et al.*, *J. Am. Chem. Soc.*, 140(45), 15384–15392, (2018)], similar DNA architectures were modified with Nitritotriacetic acid (NTA) at the locations of domain 'y. We demonstrated that in the presence of Nickel ions, Ni-NTA complexes could (reversibly) capture 6x-histidine tagged GFP. SAXS data confirmed that neither the NTA modification nor the GFP embedding impacted the crystalline structure of the aggregates. It is thus reasonable to expect that modifications of similar size applied to domain 'y should not impact the structure or stability of the particles studied in the present contribution.

We have now clarified this last point in the caption of Fig. 1:

“In the tested design, domain 'y is a non-interacting poly-T domain, which could however be replaced with a functional moiety or aptamer useful for anchoring molecular cargoes **without impacting C-star self-assembly, as demonstrated in Ref. [Brady R. A. *et al.*, *J. Am. Chem. Soc.*, 140(45), 15384–15392, (2018)]**”

Reviewer Point 1.2 — Fig S2 includes DLS values in number. Authors should also provide the values obtained in intensity.

Response: The revised version of manuscript includes an updated Fig. S2 (also included below), which now shows both the number and intensity values of the DLS signal. While the intensity distributions are naturally shifted compared to the number distributions, the relative sizes of the constructs follow the same trends, and still robustly support our interpretation of the DLS data.

Figure S2: **Dynamic light scattering of non-functionalized DNA motifs.** A shift to larger hydrodynamic diameters is observed in samples featuring motifs that are expected to connect in larger complexes (compare with AGE in Fig. S1). The hydrodynamic diameter was calculated from the averaged **number** signal of three data runs of twelve measurements each.

Reviewer Point 1.3 — Authors prepare these particles utilising DNA oligonucleotides at a micromolar concentration. Have authors explored the concentration range at which these particles are formed? Is it possible to scale up their production at larger volumes? The size of the DNA particles reaches a plateau after 50 hours at room temperature. Can authors comment on their colloidal stability on the order of days? How crucial is temperature in maintaining stability? All these features are important aspects which should be discussed specially considering potential future applications that according to authors these materials can bring as smart drug delivery systems.

Response: At the early stages of this project we have carried out optimisation of the overall DNA concentration as well as the relative ratios between the different components. The protocols presented here are the results of this optimisation. We have observed, for instance, that increasing the concentration ratio between core and corona strands by a factor 3.5 or more with respect to our established protocol leads to instability of the formed particles against coalescence, likely due to the fact that the hydrophilic corona is not sufficiently dense to offer good level of steric repulsion.

Regarding the possibility of scaling up the production to larger volumes, the only physical limitation one may encounter is the need for performing a rapid quenches, from 90°C to 65°C, and then to 35°C. Our Peltier device can only achieve sufficiently fast rates for relatively small samples, hence our choice of preparing samples in small-volume glass capillaries. However, there would be no intrinsic technical challenges in constructing a more powerful Peltier device with better optimised heat exchange, and/or one that can host a large number of small-volume samples.

The particles remain stable for several weeks after production. To prove this, in revised Fig. S7 (also included below) we have included DDM-determined particle sizes obtained after 8 and 14 days from assembly. Size remained unchanged, demonstrating long-term stability.

If left in the sealed capillaries in which thermal processing is carried out, particles can be stored long term at room temperature, thanks to the sterilising and enzyme-deactivating action of the thermal processing. Samples used for long-term stability assessment (Fig. S7) were stored this way. If the particles are extracted from the capillaries and sterility compromised, then cooling would be beneficial to slow down degradation from possible biological contaminants.

In the revised version of the main text we have commented about the long-term stability of the particles with reference to Fig. S7:

“Notably, particle samples can be stored long-term at room temperature, and retain their colloidal stability for at least 14 days post assembly (Fig. S7).”

Figure S7: **Long-term stability of protected particles as determined with DDM.** After a slight growth, hydrodynamic radius of particles as determined with DDM stabilizes, proving their stability against coalescence. PH was measured for 50 h (3000 minutes) at room temperature. Two additional measurements were taken after 8 and 14 days for the sample assembled with $\tau_g = 1800$ s. Compare with the curves collected for all tested growth times at shorter wait times in Fig. 2b.

Reviewer Point 1.4 — Large particles cannot be observed by TEM and authors indicate this may be due to TEM sample preparation. Authors should consider alternative ways to image them such as AFM.

Response: The instability that prevented us to properly image larger particles in TEM was due to the detrimental impact of the low-pH uranyl acetate solution the samples needed to be exposed to for negative-staining.

For this revision, we have repeated all TEM measurements without such staining, with a much gentler sample preparation protocol that does not disrupt large particles. The new measurements, summarised in revised Figure S9 (also shown below), resulted in particles sizes aligned with those determined with DDM over a broad range of growth times.

In the revised manuscript we have modified the TEM section in the Methods to reflect the new protocol, and updated the relevant statement in the main text:

"Particles prepared at various τ_g have been imaged with TEM, and found to possess a roughly spherical morphology (Fig. 2c) and sizes aligned with those determined by DDM (Fig. S9)"

Figure S9: **Size distribution of protected particles as a function of growth time as determined with TEM.** For all the studied growth times, the particle-size dependency on τ_g follows a monotonic increase, which is consistent with the trend determined by DDM (Fig. 2b) and DLS (Fig. S6). The measured radii are in agreement with the values obtained by DDM. Each data-point represents a single particle whose size was determined by taking four intensity profiles along the center of the particle, calculating a derivative of the obtained signal and measuring a distance between two local maxima corresponding to a border of the particle. Top: selected TEM micrographs from each sample. Scale bars are 200 nm.

Reviewer Point 1.5 — In figure 4c and d, the effect of DNA particle concentration on GUVs disruption and calcein leakage is shown. Fig S12 shows this DNA particle concentration follows Hill function. The molar concentration of GUVs has also a direct effect on these dose-dependent “lethality” but it is not included in the text. Likewise, LUVs concentration is not shown. Authors should include these data.

Response: In our revised contribution we indicated the molar concentration of lipids in our GUV and LUV samples, as determined using an enzymatic quantitation assay described in the revised Methods section. For non-tagged and fluorescently tagged GUVs we obtained $3.12 \pm 0.16 \text{ g } \Lambda^{-1}$ and $1.66 \pm 0.14 \text{ g } \Lambda^{-1}$, respectively. For LUVs we obtained $0.96 \pm 0.12 \text{ g } \Lambda^{-1}$. Lipid concentration in GUV and LUV samples is now reported in the relevant Methods sections and the caption of relevant Figures (4, S13, S15). The horizontal axis of Fig. S14 has been updated to report the DNA/lipid molar ratio, so to better represent DNA “dose”.

Reviewer 2

In this manuscript, DNA hydrogel particles are developed that contain a hydrophobic cholesterol functionality. These cholesterol groups are initially protected at the surface of the particle by capping DNA structures that can be removed by toe-hold strand displacement by an additional DNA “trigger”. Once deprotected, these hydrophobic hydrogel particles further aggregate into larger gel structures and/or interact with coexisting lipid vesicles in their surroundings. This destabilises the GUVs causing them to leak and collapse - a common effect when GUVs are subjected to a high concentration of membrane-active substances. Some GUVs also become encased in a complete gel corona such that they arrest in state. None of these results are particularly unexpected or surprising. The paper is clearly written and the experiments appear to be well performed, analysed and presented. The effect of these hydrogel particles on GUVs is clear. What I do struggle with is the “why?”. This is something the authors should aim to expand on and clarify for the paper to be of broad interest. At present I cannot put my finger on either the new fundamental scientific understanding that is gained or the exciting new application that this work enables. It currently falls a little into the no-man’s-land between the two, achieving neither. I find the analogies with natural biological processes tenuous and the work does not add any new insight to these. Similarly the potential applications in biosensors, therapeutic technologies or synthetic biology are highly speculative and none are well borne out such as to become convincing as real or viable. Perhaps focus on one of these, fleshing out the details and a proof of concept demonstration would significantly strengthen the interest of the paper for a broad multidisciplinary audience.

Response: We thank the reviewer for the constructive criticism, which prompted us to explore experimentally one of the foreseen bio-inspired functionalities of our system, in an effort to broaden the interest of our contribution.

In our initial submission, we had mentioned that the ability of triggered DNA particles to embed cell-sized GUVs resembles that of the DNA networks secreted by certain immune cells (neutrophils) to trap pathogens. These structures, known as Neutrophils Extracellular Traps (NETs) are an important component of the innate immune response that has been thoroughly investigated in biology.

In the resubmitted manuscript we have directly explored this potential analogy with a new series of experiments aimed at ascertaining the ability of the network-forming DNA particles to arrest the motion of model swimming bacteria. The results are summarised in new Fig. 5, Figs S16 and S17, and discussed

in a new section in the main text.

Briefly, we have incubated samples of an *E. coli* strain typically used for motility studies (MG1655) with particles with and without the addition of the trigger oligonucleotide inducing their aggregation, following a protocol similar to the one applied for GUV destabilisation experiments.

Confocal microscopy images (Fig. 5b) prove that the bacteria (red) indeed become stuck to, or embedded within the DNA networks formed in the triggered samples (cyan), while remaining uniformly distributed if particle aggregation is not initiated. For quantitative assessment of the impact of DNA-network formation on motility we have defined a "motility parameter", extracted from bright field microscopy videos (see definition in main text and Methods). Frame-wide maps of this parameter in Fig. 5c highlight a substantially lower degree of motion in the sample with aggregating particles at the final time point of our experiment. The time evolution of the motility parameter, summarised in Fig. 5d (frame-averaged parameter, see Fig. S16 for the full colormaps), proves that while in both triggered and un-triggered samples an initial increase in bacteria motion is observed, a plateau is quickly reached in the presence of particle aggregation, following bacteria entrapment. The increase in motility is a consequence of bacteria replication, expected in both samples due to the known ability of *E. coli* to use nucleic acids as a food source [Finkel S. E. *et al.*, *J. Bacteriol.*, 183(21), 6288-6293, (2001)], as confirmed by data in Fig. 5e, reporting the emission intensity of a fluorescent protein (mKate2) expressed by *E. coli* (see also Fig. S17 for turbidity data). Here, curves for both triggered and un-triggered particles display a similar steady growth throughout the experiment, demonstrating that the number of bacteria ramps up similarly in both samples. We can thus safely conclude that the marked difference in the motility parameter shown in Fig. 5c, d and Fig. S16 is indeed a consequence of the ability of the DNA-particle network to immobilise bacteria, rather than following from difference in *E. coli* numbers.

We believe that this entirely new section of the manuscript clearly demonstrates a biology-inspired application of our particles, that could potentially be developed into anti-microbial technologies but also directly appeal to the rapidly expanding community interested in replicating complex life-like responses in synthetic systems. Notably, to the best of our knowledge, this is the first example of a biomimetic implementation that reproduces the effect of neurophil-secreted DNA NETs. We hope that the Reviewer agrees with us on the fact that this addition substantially broadens the interest and potential impact of our contribution.

Alongside the new main-text section and aforementioned figures, the manuscript has been updated in all key sections, including title, abstract, conclusion and Methods to reflect the new content. We do not quote the modified text here for brevity, but invite the Reviewer to refer to the marked version of the main text and SI where all changes are highlighted in red.

Reviewer 3

Walczak et al. assembled DNA-based particles with amphiphilic core and hydrophilic shell from short single-stranded DNA molecules. The particles were assembled in one-pot reaction with three temperature steps. By increasing the incubation time at 65° C, particles with bigger sizes were assembled. The particles were characterized with differential dynamic microscopy and fluorescence microscopy. Notably, the hydrophilic shell can be displaced by a strand-displacing reaction, leading

to destabilization of the particles. These deprotected particles showed enhanced rupturing activity of GUVs. Although the one-pot assembly of core-shelled particle out of oligonucleotides is impressive, I do not feel the claims in the manuscript are justified by the data presented in the manuscript. Especially, high amount of leaks in the figure 4b concerns me. Also, the protected particles grow significantly in terms of their volume over time (Figure S7) and shows significant activity for rupturing GUVs (Figure 4a). Therefore, I believe the stability and activity of the DNA particles should be improved before it can be published in Nature Communications. I have put more detailed moments below.

Response: We thank the reviewer for the insightful assessment. We have now carried out additional tests to further demonstrate the stability and functionality of our particles, which we discuss below in a point-by-point response to the concerns raised.

Reviewer Point 3.1 In figure 2a, particles are difficult to see. Please consider enlarging the snapshot and reducing the field of view.

Response: We agree with the Reviewer that particles were rather difficult to see in the original Fig. 2a. We have now replaced the micrographs with new ones relative to a smaller field of view, as well as adjusting the contrast (as now specified in the caption). The particles can now be seen clearly for the larger t_g values, but we do not expect them to be discernible by eye for smaller incubation times, given their small (sub-diffraction) size. The objective of these snapshots is indeed that of giving the reader a visual appreciation of the t_g -dependent particle size quantitatively assessed in the plot below and other SI data (DLS, TEM). Note finally that the fact that the smaller particles cannot be individually resolved does not impact the ability of DDM to detect their size quantitatively, as demonstrated in several instances in literature [Safari M. S. *et al.*, *npj Microgravity*, 3, 21, (2017); Ferri F. *et al.*, *Eur. Phys. J. Spec. Top.*, 199, 139–148, (2011)].

Reviewer Point 3.2 I do not completely agree with the stability argument presented in figure 2b. The particles grow 10–20% in terms of hydrodynamic radius within one minute. Compared to the growth curve presented in Figure 2a, it is still significant and fast. There are some long-term stability curves in Figure S7, but it should be investigated further and incorporated in the protocol for figure 4. The particles might behave better if they are incubated for ~ 400 min before rupturing experiments. Also, considering the hydrodynamic radius $rh \propto V^{1/3}$, the change in volume could be significant over time.

Response: First, we noticed that the x-axis of Fig. 2b in our initial submission contained a typo: the units of time should have been minutes, rather than seconds. We have now fixed this mistake. Therefore, the slight growth the Reviewer refers to occurs over one hour rather than one minute, and thus is substantially slower compared to the growth times considered in Fig. 2a. We apologise for the mistake and the confusion it created.

We can thus safely claim that, particularly for the small-size particles used in membrane rupture and the newly included bacteria trapping experiments, the vast majority of the growth occurs during the dedicated growth phase at $T = 65^\circ\text{C}$, and then proceeds at a comparatively negligible rate after the temperature has been further quenched to $T = 35^\circ\text{C}$, inducing corona formation. We would also like to point out that the apparent residual growth observed in Fig. 2b, particularly for larger particles, may also be an artefact following from the equilibration of the barometric distribution of the particles, with the larger ones among the (naturally polydisperse, see Fig. S5) population concentrating close to the

bottom of the chamber where the DDM videos are collected.

Sizes plotted vs τ_g in Fig. 2a are those recorded after wait times of 60 minutes from the end of the growth stage, *i.e.* the longest delay time in Fig. 2b. This delay time was chosen to allow the residual growth (or barometric equilibration) to reach a plateau, which appears to be approached for many of the curves in Fig. 2b. In the revised SI (Fig. S8, also shown below) we also include an alternative version of the plots in Fig. 2a, in which rather than the $\tau = 60$ min point we considered sizes averaged over all the delay times in Fig. 2b. The two versions of the growth curves are in good quantitative agreement if one accounts for the relatively large errorbars in the measured sizes, indicating that the residual growth has no impact on our claim of programmable particle size supported by Fig. 2a.

Regarding the incubation/delay time elapsed between particle preparation and rupture/trapping experiments: in all cases the particles were stored for at least 1 day (at room temperature) before being used in these experiments. Note that, and as discussed in detail when addressing point 3 of Reviewer 1, the particles remain stable for several days after production, as further demonstrated by the newly-included size measurements performed in samples stored for 8 and 14 days (revised Fig. S7, shown above, point 3 of Reviewer 1).

In the revised main text, besides fixing the typo in the x-axis label of Fig. 2b, we clarified our argument regarding the residual growth:

“This slight post-assembly growth can be a consequence of a small degree of further particle coalescence or, for larger sizes, equilibration of the barometric particle distribution.”

Additionally we comment on the new long-term stability data:

“Notably, particle samples can be stored long-term at room temperature, and retain their colloidal stability for at least 14 days post assembly (Fig. S7).”

2000

Figure S8: **Hydrodynamic radius of protected particles with displaceable and non-displaceable corona.** Data analogous to those shown in Fig. 2a, but in which the hydrodynamic radius is averaged over the time interval covered in Fig. 2b, rather than just considering the endpoint of the experiment (60 minutes). Note the very similar trends compared to those in Fig. 2a. Dashed lines are best fits to a diffusion-reaction growth model as summarised in Supplementary Discussion 1. Parameters: $A = 2.57 \times 10^{13} \text{ s m}^{-3}$, $B = 1.11 \times 10^{15} \text{ s m}^{-2}$, $X = -13.5 \text{ s}$, and $A = 2.42 \times 10^{13} \text{ s m}^{-3}$, $B = 1.40 \times 10^{15} \text{ s m}^{-2}$, $X = -11.35 \text{ s}$, for non-displaceable and displaceable, respectively.

Reviewer Point 3.3 — It is difficult to see and therefore claim a monotonic increase in Figure S8. If the authors still want to use TEM, please consider CryoTEM. But I do not feel a TEM is necessary.

Response: The unexpected trend in the TEM data observed in the initially submitted manuscript was due to sample de-stabilisation, induced in turn by the low-pH uranyl acetate solution needed for negative staining. For this resubmission, we have repeated all TEM measurements without applying any staining, so to eliminate the harsh processing steps that caused de-stabilisation, and relying only on the natural contrast from the dense DNA particles. The results, summarised in the revised Fig. S9 (shown above, point 4 of Reviewer 1) show a clear monotonic increase of the particle size with incubation time, as well

as good agreement of the measured size with those obtained via DDM (Fig. 2a).

We have considered and attempted cryoEM, but in some conditions particles are too large to be embedded and guarantee an optimal thickness of the vitreous ice, hence our decision to optimise regular TEM.

In the revised manuscript we have modified the TEM section in the Methods to reflect the new protocol, and updated the relevant statement in the main text:

“Particles prepared at various τ_g have been imaged with TEM, and found to possess a roughly spherical morphology (Fig. 2c) and sizes aligned with those determined by DDM (Fig. S9).”

Reviewer Point 3.4 — I am not sure the fluorescence intensity on surface could serve as a quantitative marker for aggregation in Figure 3b. It clearly shows kinetics involving aggregation and sinking of the particles which is difficult to correlate directly with growth. I suggest either use mean fluorescence intensity of particles or develop a mathematical model for the data.

Response: We understand the concern of the Reviewer on this point, and agree that our approach based on recorded fluorescence intensity cannot give quantitative information on the aggregation kinetics of the particles. In fact, data in Fig. 3b were primarily aimed at demonstrating that trigger addition leads to particle destabilisation, and to provide only a rough estimate of the timescales involved. As noted by the reviewer, the fluorescence intensity curve acquired is influenced by sedimentation. However, sedimentation only occurs following particle aggregation and the formation of larger, heavier objects. Stable particles of the relevant size ($P_H \sim 500$ nm) do not display any sedimentation over time, as we demonstrate with newly included data on un-triggered particles (see the relevant plot below). Here, the fluorescence trace remains low and perfectly constant over the duration of the experiment, demonstrating that the steady increase observed in the presence of trigger follows indeed from triggered aggregation and the consequent sedimentation.

Following the Reviewer’s suggestion, we have explored the extensive classic literature on modelling colloidal aggregation coupled with sedimentation. Owing to the complexity of these phenomena, that in different regimes are influenced by factors such as hydrodynamics and mechanical stability of aggregates, there are not, to the best of our knowledge, tractable expressions that could be used to extract quantitative information from our fluorescent traces. It should also be noted that, while most classic studies refer to solid colloidal particles, in our situation, once the DNA-particles are unprotected they undergo partial coalescence, and the resulting aggregates coarsen visibly over time. The fluorescent signal recorded at the bottom will thus also be influenced by these structural rearrangements.

In the revised Fig. 3b we have included the aforementioned control trace, further strengthening the claim that the increase in fluorescence in the triggered samples is a good indicator of particle aggregation. Methods and caption have been updated accordingly.

Figure 3b: **Triggered release of protective corona leads to particle aggregation.** Corona displacement leads to the exposure of the ‘sticky’ C-star core and subsequent particle aggregation, assessed by measuring the time-dependent hydrodynamic radius as determined *via* DDM (blue circles) and the normalized fluorescence intensity of labelled core motifs (red circles) after the addition of trigger strands ($\tau = 0$). **The increase in R_H observed upon trigger addition follows from the formation of larger aggregates, while the increase in the fluorescence trace is caused by their progressive sedimentation at the bottom of the cell, where the signal is recorded.** For both observables, data are shown as mean \pm standard deviation of 3 independent repeats. **Red triangles indicate a control fluorescent trace measured in the absence of trigger. The constant and low value confirms the absence of spontaneous sedimentation.** Top: bright field and fluorescence micrographs at different time-points after the addition of the trigger ($\tau = 100, 300, 500$ and 690 min). All scale bars $25 \mu\text{m}$.

Reviewer Point 3.5 Figure parts labelling should be a, b rather than a. b.

Response: We thank the Reviewer for picking up this editing issue. We have fixed it by changing the panel labels.

Reviewer Point 3.6 The labelling scheme of lines in Figure 4c and d could be improved. Using the same colour and shape combination to mark different conditions and concentrations at the same time could confuse readers.

Response: We agree that the previous labeling scheme of lines in Fig. 4c, d was confusing. We fixed this issue by implementing a new labeling system with a different combination of shape and colour representation for each of the experimental conditions.

Reviewer Point 3.7 The Figure 4 should be the highlight of the manuscript but it is bit worrying. First of all, calcein should not leak from GUV in a normal condition. In the Figure 4b, the GUV without any trigger releases calcein at 100 minutes timescale. However, the preparation of the GUV and chamber should take hours. Then why the GUVs still have calcein inside after hours of preparation? This suggest the GUVs in the microfluidic chamber is somehow destabilized. The destabilization could come from difference in ionic strength or high density of biotin/streptavidin on the membrane/surface. Also, if the authors find it difficult to prevent leakage of calcein, other dyes such as sulforhodamine B can be used as well.

Response: We thank the Reviewer for this comment that prompted us to repeat the measurement in question using a different fluorescent cargo. We opted for fluorescein-sodium, as having used it in the past we noticed small leakage rate from vesicles. The new data, summarised in Fig. 4d (bottom left, also included below), shows a much reduced rate of fluorescence decrease in sample with DNA particles, but no trigger, and in those without particles, compared with the case in which both particles and aggregation-trigger are included. The residual fluorescent reduction is likely ascribable to bleaching, rather than leakage. In the revised manuscript the Methods have been updated to describe the change in experimental protocol.

Figure 4d: **Unprotected particles trigger vesicle rupture and cargo release.** Progressive leakage of fluorescein-sodium initially encapsulated in GUVs as quantified with confocal microscopy. Bottom: unprotected particles significantly increase the spontaneous leakage rate compared to control samples of unperturbed GUVs and those exposed to stabilized DNA particles. Top: Confocal micrographs demonstrating fluorescein-sodium (cyan) leakage following the adhesion of DNA particles (TXRED, red) onto GUVs. Scale bars 5 μ m. **The (mass) concentration of GUVs was 3.12 ± 0.16 g L $^{-1}$.**

Reviewer Point 3.8 — Is there any reading to use the particles grown for 15 seconds to rupture the membrane? Have authors tried particles with bigger sizes to rupture the membrane?

Response: The decision to go for some of the smallest particles for our membrane rupture experiments was motivated by their greater diffusion coefficient and less pronounced tendency to sediment. Once the protective corona is removed, besides adhering to nearby membranes, particles will also start aggregating with each other. As they grow larger, aggregates will then start sedimenting, and eventually will settle at the bottom of the experimental chamber with little or no residual Brownian diffusion. If by this stage, particle aggregates have not yet encountered a GUV, they will be highly unlikely to produce any disruption. If particles start off smaller, the initially formed “sticky” aggregates will also be smaller and more diffusive, hence increasing the chances of them encountering a GUV before large-scale aggregation and sedimentation occur. To confirm this hypothesis, in the revised submission we have carried out membrane-disruption experiments with larger particles, namely those produced with $\tau_g = 180$ s and $\tau_g = 600$ s. Consistent with our interpretation, the results summarised in the newly included Fig. S13

(also included below) indicate a less pronounced ability of these particles to cause GUV rupture.

In the revised main text we have included a summary of this discussion and referred to the new data in Fig. S13.

“Consistent with this picture, Fig. S13 demonstrates that larger particles, expected to rapidly form sedimenting and slowly diffusing aggregates upon trigger addition, are less efficient in disrupting GUVs compared to the smaller particles examined in Fig. 4c.”

Figure S13: **GUV rupture induced by particles of larger sizes.** Particles incubated for growth times (τ_g) longer than 15 s have a significantly less disruptive influence on the membrane stability compared to their smaller counterparts (see Fig. 4c). This can be ascribed to the reduced diffusivity of larger particles, which leads to a limited DNA aggregation on GUVs surface at the initial stage of the experiment when membrane rupture is most prominent. The fraction of “surviving” GUVs was calculated from confocal micrographs taking the number of vesicles present in the field of view at $\tau = 0$ as reference. The (mass) concentration of GUVs was $3.12 \pm 0.16 \text{ g L}^{-1}$.

REVIEWER COMMENTS

Reviewer #1 (Remarks to the Author):

Authors have carried out a thorough revision of the manuscript and after having addressed the points raised by referees the paper has nicely improved. Just as a minor point, authors should include as a comment (either in the methods part or supporting), their experimental observation regarding the importance of keeping the concentration ratio between core and corona strands by a factor below 3.5. This piece of information can be helpful for other authors in future to prepare these particles. Apart from this minor comment, I consider the manuscript is suitable for publication in Nature Communications in its current form.

Reviewer #2 (Remarks to the Author):

The authors have made significant efforts to address my criticism of a lack of clear demonstrated application by investigating the capture of bacteria in DNA nets. This adds novelty and interest to the work. To better link between the GUV and bacteria sections of the work, it would be useful to consider the impact of the nanoparticles on the leakage of E coli membranes. Do the particles have a similar effect in damaging the integrity of the membrane or not?

Reviewer #3 (Remarks to the Author):

I am happy with the responses from the authors.
I recommend publishing the manuscript as it is.

Author's Response to the Reviewers

Reviewer 1

Authors have carried out a thorough revision of the manuscript and after having addressed the points raised by referees the paper has nicely improved. Just as a minor point, authors should include as a comment (either in the methods part or supporting), their experimental observation regarding the importance of keeping the concentration ratio between core and corona strands by a factor below 3.5. This piece of information can be helpful for other authors in future to prepare these particles. Apart from this minor comment, I consider the manuscript is suitable for publication in Nature Communications in its current form.

Response: We thank the reviewer for pointing out that we had missed this important piece of information. We have now included the following sentence in the methods:

Note that changing oligonucleotide concentrations with respect to the values reported in SI Table 2 may lead to reduced particle stability. In particular, we have observed that stability against aggregation is compromised if the concentration ratio between core and corona strands is increased by a factor 3.5 or more with respect to the reported protocol.

Reviewer 2

The authors have made significant efforts to address my criticism of a lack of clear demonstrated application by investigating the capture of bacteria in DNA nets. This adds novelty and interest to the work. To better link between the GUV and bacteria sections of the work, it would be useful to consider the impact of the nanoparticles on the leakage of E coli membranes. Do the particles have a similar effect in damaging the integrity of the membrane or not?

Response: We agree with the Reviewer that this point deserved further discussion. The evidence that bacteria continue to grow in the presence of network-forming DNA particles suggests that the latter, while successfully trapping the bacteria and arresting their motion, does not significantly damage them. We would indeed expect that if the DNA aggregates were to trigger a substantial leakage in the *E. coli* cell-wall, then the bacteria would not display the substantial growth we observe. To further confirm this hypothesis we inspected our epi-fluorescence time-lapse data to specifically monitor the growth and structural integrity of a few individual bacterial cells embedded in the DNA network. Examples shown in the newly included Fig. S18 confirm that the cells retain their regular morphology and growth behaviour, thus ruling out substantial de-stabilisation of the cell-wall. In addition, we note the absence of a progressive decrease of the mKate fluorescent signal from the bacterial cells, which one would expect if the cell walls had become substantially damaged and leaky to the fluorescent protein.

It is not surprising that the amphiphilic DNA particles do not have on the bacterial cell-wall the same destabilising effect observed in bare bilayers. Indeed, both the rigid peptidoglycan layer underlying the outer lipid membrane, and in particular the lipopolysaccharide brush surrounding the latter are expected to provide enhanced resilience.

Besides including the new SI Figure, we have commented on these points in the revised main text.

Detailed inspection of the microscopy images, exemplified in Fig. S18, confirms that bacteria can grow while embedded in the amphiphilic DNA network, and retain their structural integrity in this environment. The latter point indicates a greater robustness of the cell-wall against de-stabilisation by the amphiphilic DNA network compared to bare lipid membranes, likely as a consequence of presence of the protective lipopolysaccharide layer [Madigan M. T. *et al.*, *Brock Biology of Microorganisms*, Pearson, pp. 75-84 (2018)].

Figure S18: ***E. coli* grow while embedded in the amphiphilic DNA network.** Epifluorescence micrographs demonstrating that *E. coli* embedded in the amphiphilic DNA network retain their normal morphology, continue to grow, and do not lose mKate2 fluorescent signal through leakage. This indicates that, while successfully immobilising the cells, the DNA network does not significantly destabilise the bacterial cell wall, which could be a consequence of the protective action of the lipopolysaccharide layer surrounding the outer membrane. The trigger strand was added at time 0. Core motifs are shown in cyan (fluorescein), *E. coli* in red (mKate2). Scale bar 3 μm .

REVIEWERS' COMMENTS

Reviewer #2 (Remarks to the Author):

The authors have comprehensively addressed my questions and this is now an excellent contribution that I recommend for publication in Nature Communications.

There is one minor error in the Fig S18 caption: "lose" instead of "loose".